# Differential Transcriptomic Profiles Following Stimulation with Lipopolysaccharide in Intestinal Organoids from Dogs with Inflammatory Bowel Disease and Intestinal Mast Cell Tumor

**DOI:** 10.3390/cancers14143525

**Published:** 2022-07-20

**Authors:** Dipak Kumar Sahoo, Dana C. Borcherding, Lawrance Chandra, Albert E. Jergens, Todd Atherly, Agnes Bourgois-Mochel, N. Matthew Ellinwood, Elizabeth Snella, Andrew J. Severin, Martin Martin, Karin Allenspach, Jonathan P. Mochel

**Affiliations:** 1Department of Veterinary Clinical Sciences, College of Veterinary Medicine, Iowa State University, Ames, IA 50011, USA; bdana@wustl.edu (D.C.B.); lawrancecc@gmail.com (L.C.); ajergens@iastate.edu (A.E.J.); atherlyt@gmail.com (T.A.); abmochel@iastate.edu (A.B.-M.); allek@iastate.edu (K.A.); 2SMART Pharmacology, Department of Biomedical Sciences, College of Veterinary Medicine, Iowa State University, Ames, IA 50011, USA; 3Department of Animal Science, Iowa State University, Ames, IA 50011, USA; matthew@mpssociety.org (N.M.E.); esnella@iastate.edu (E.S.); 4Office of Biotechnology’s Genome Informatics Facility, Iowa State University, Ames, IA 50011, USA; severin@iastate.edu; 5Mayo Clinic, Rochester, MN 55905, USA; mmartin@mednet.ucla.edu

**Keywords:** LPS, enteroids, colonoids, IBD, mast cell tumor, canine, microarray

## Abstract

**Simple Summary:**

Lipopolysaccharide (LPS) derived from intestinal bacteria is linked to long-lasting inflammation that contributes to the development of intestinal cancer. While much research has been performed on the interplay between LPS and intestinal immune cells, little is known about how LPS influences intestinal epithelial cell structure and function. In this study, we investigated the effects of LPS on the proliferation and function of genes in intestinal organoids derived from dogs with gastrointestinal diseases, including inflammatory bowel disease (IBD) and intestinal mast cell tumor. The goal of this study was to evaluate how LPS affects signaling pathways in intestinal epithelial cells to influence development of a pro-tumor-like environment. Using an ex vivo model system, LPS incubation of organoids activated cancer-causing genes and accelerated the formation of IBD organoids derived from the small and large intestines. In brief, the crosstalk that occurs between the LPS/TLR4 signal transduction pathway and several different metabolic pathways, including primary bile acid biosynthesis and secretion, peroxisome, renin-angiotensin system, glutathione metabolism, and arachidonic acid pathways, may play a prominent role in the development of chronic intestinal inflammation and intestinal cancer.

**Abstract:**

Lipopolysaccharide (LPS) is associated with chronic intestinal inflammation and promotes intestinal cancer progression in the gut. While the interplay between LPS and intestinal immune cells has been well-characterized, little is known about LPS and the intestinal epithelium interactions. In this study, we explored the differential effects of LPS on proliferation and the transcriptome in 3D enteroids/colonoids obtained from dogs with naturally occurring gastrointestinal (GI) diseases including inflammatory bowel disease (IBD) and intestinal mast cell tumor. The study objective was to analyze the LPS-induced modulation of signaling pathways involving the intestinal epithelia and contributing to colorectal cancer development in the context of an inflammatory (IBD) or a tumor microenvironment. While LPS incubation resulted in a pro-cancer gene expression pattern and stimulated proliferation of IBD enteroids and colonoids, downregulation of several cancer-associated genes such as *Gpatch4*, *SLC7A1*, *ATP13A2*, and *TEX45* was also observed in tumor enteroids. Genes participating in porphyrin metabolism (*CP*), nucleocytoplasmic transport (*EEF1A1*), arachidonic acid, and glutathione metabolism (*GPX1*) exhibited a similar pattern of altered expression between IBD enteroids and IBD colonoids following LPS stimulation. In contrast, genes involved in anion transport, transcription and translation, apoptotic processes, and regulation of adaptive immune responses showed the opposite expression patterns between IBD enteroids and colonoids following LPS treatment. In brief, the crosstalk between LPS/TLR4 signal transduction pathway and several metabolic pathways such as primary bile acid biosynthesis and secretion, peroxisome, renin–angiotensin system, glutathione metabolism, and arachidonic acid pathways may be important in driving chronic intestinal inflammation and intestinal carcinogenesis.

## 1. Introduction

Lipopolysaccharide (LPS) is the most effective cell wall-derived inflammatory toxin of Gram-negative bacteria with its inner component, lipid A, accountable for most of the toxin’s inflammatory effects [1]. The intestinal lumen, a habitat for many trillions of commensal bacteria, is the primary LPS reservoir in the body [2]. During normal circumstances, the intestinal epithelium retains a proper barrier function by promoting the transcellular movement of nutrients, water, and ions while reducing the paracellular transport of bacteria or its products/toxins (including LPS) into the systemic circulation [3]. In intestinal permeability disorders such as infection by pathogenic bacteria and faulty detoxification of LPS, the defective tight junction (TJ) barrier will permit the enhanced paracellular flux of LPS and other phlogistic luminal antigens [4]. Interestingly, apical but not basolateral exposure to LPS, induces epithelial apoptosis through caspase-3 activation and stimulates the disruption of tight junctional ZO-1, thus increasing epithelial permeability [5]. While transported in blood, LPS binds to either the LPS binding protein (LBP) or plasma lipoproteins and induces systemic inflammation [4]. Basically, LPS-induced intestinal inflammation occurs through the stimulation of Toll-like receptor 4 (TLR4), which subsequently leads to the recruitment of intracellular nuclear transcription factor-κB (NF-κB), followed by the release of chemokines and inflammatory cytokines including necrosis factor-alpha (TNF) [6,7]. TNF is accountable for various signaling events within cells, leading to necrosis or apoptosis, therefore playing a pivotal role in the resistance to infection and cancers. Activation of the TLR-4 dependent FAK-MyD88-IRAK4 signaling pathway controls LPS-induced intestinal inflammation and tight junction permeability [8,9].

The removal of circulating LPS (via amelioration of dysbiosis) facilitates the clinical recovery of inflammatory bowel disease (IBD), demonstrating its significant role in mediating chronic intestinal inflammation in IBD [2]. The biological role of LPS has been extensively characterized, particularly in the context of immune cell activation, where it causes pleiotropic inflammatory cytokine and chemokine secretion [10] as well as its interactions with intestinal macrophages, dendritic cells, and T cells [11,12]. The pathogenesis of canine IBD is multifactorial and involves a loss of tolerance to diet and microbial components that cause an aberrant immune response in genetically susceptible hosts. Intestinal dysbiosis characterized by increased abundance of Gamma-Proteobacteria (e.g., *Enterobacteriaceae*) [13] and defects in innate immune sensing are pivotal factors that initiate and drive chronic intestinal inflammation.

Mast cells (MCs) play an essential immunoregulatory role, especially at the mucosal barrier between the body and the environment. In intestinal lesions, MCs can be seen infiltrating the inflammatory microenvironment and regulating the synthesis of several pro-inflammatory cytokines and mediators of inflammatory cell production, thus promoting tumor growth and proliferation [14]. Numerous animal and human studies indicate the presence of an abundance of MCs surrounding the tumor cells or inflammation lesions [14,15,16,17]. For instance, high MC density is correlated with advanced stage and tumor progression in colorectal cancer (CRC) [17]. Similarly, intratumoral MC density correlates with tumor size, and peritumoral MC density correlates with the malignancy grade [16]. MCs also play a crucial part in IBD, and research has shown that the number of MCs increases in inflammatory bowel lesions in patients with IBD [14]. As a trigger of inflammatory responses, LPS has been associated with cancer pathogenesis including the development of gastrointestinal (GI) mast cell tumors and CRC [18,19,20]. Different microbial products and TLR4 agonists have demonstrated the crucial role of TLR4 signaling in regulating tumor growth, survival, and progression in colonic, pancreatic, liver, and breast cancers [21]. The TLR4 signaling pathway has been shown to drive tumorigenesis and exhibit some antitumor effect on TLR4 activation [22,23]. TLR4 serves as a key bridge molecule linking oncogenic infection to colonic inflammatory and malignant processes by inducing miR-155 expression via transcriptional and post-transcriptional mechanisms [24]. Conversely, miR-155 can also augment TLR4 signaling by targeting negative regulators SOCS1 and SHIP1 [24]. Regardless of the significance of an impaired intestinal barrier indicated by elevated circulating LPS levels, the role of LPS in regulating signaling pathways involved in the development of intestinal inflammation in IBD remains mostly unknown. Gene expression profiling using microarray technology has been applied to identify physiologically and clinically significant subgroups of TNF-responsive tumors, elucidate the combinatorial and complex nature of cancers, [25,26] and advance our mechanistic understanding of oncogenesis.

The dog genome and its organization have been studied extensively in the last ten years. To understand the biology of human diseases, dogs have emerged as a primary large animal model [27,28,29,30,31]. Of the large animal models used in translational GI research, the dog is particularly relevant since it presents similar gut physiology, dietary habits, and intestinal microbiota to humans [29,32]. Moreover, dogs spontaneously acquire severe chronic intestinal diseases such as IBD and CRC, which make them ideal animal models for translational GI research [31]. Our laboratory successfully developed and characterized a canine 3D enteroids/colonoids model system to study intestinal biology during health and disease and bridge the translational knowledge gap between mice and humans [30,33]. Adult stem cells can differentiate into different intestinal epithelial cells, so that the epithelial cell types normally present in vivo are also identified in cultured intestinal organoids. Adult stem cell-derived intestinal organoids have previously been shown to retain their genetic and epigenetic phenotype in vitro after numerous passages [34]. We, therefore, hypothesized that a stromal mast cell tumor in the small intestine of a dog could influence the phenotypic expression of the overlying epithelium, which would be retained in the organoid culture. Mast cell tumors are known to secret histamine, proteases, prostaglandin D2, leukotrienes, and heparin as well as a variety of pro-inflammatory cytokines [35], which could change the expression profile of the epithelium overlying the tumor and sensitize it to the effects of LPS. The current study aimed to broaden our understanding of the LPS-induced regulation of signaling pathways in the intestinal epithelium and identify novel genes involved in inflammatory disease and colorectal cancer development.

## 2. Materials and Methods

### 2.1. Ethical Animal Use

The collection and analysis of intestinal biopsy samples from dogs with IBD and mast cell tumors were previously approved by the Iowa State University (ISU) Institutional Animal Care and Use Committee (IACUC-19-102; PI: Albert E. Jergens). All methods were performed in accordance with the relevant guidelines and regulations of IACUC as required by U.S. federal regulations [36]. The study is reported in accordance with the ARRIVE guidelines (https://arriveguidelines.org, accessed on 29 November 2018).

### 2.2. Crypt Cell Isolation and Enrichment for Enteroid and Colonoid Culture

Ten to fifteen endoscopic mucosal biopsies (using 2.8/3.2 mm forceps cup) of the ileum and colon were obtained from two dogs, one diagnosed with IBD (to derive colonoids) and another one with a mast cell tumor (to derive enteroids/colonoids). Epithelial crypts were isolated and enriched from intestinal biopsy samples, as reported previously [37]. Briefly, ileal and colonic biopsies were cut into small pieces (1–2 mm thickness) with a scalpel and were washed six times using the complete chelating solution (1X CCS) (Table 1). Pre-wetted pipette and conical tubes with 1% bovine serum albumin (BSA) were used throughout the procedure to prevent the adherence of the crypt epithelia in the tubes and pipette, thereby minimizing the loss of cryptal units [37]. Then, tissues were incubated with 1X CCS containing EDTA (20 mM–30 mM) for 45 min for colonic biopsies and 75 min for ileal biopsies at 4 °C on a 20-degree, 24 rpm mixer/rocker (ThermoFisher Scientific, Rochester, NY, USA). After EDTA chelation, the cryptal epithelia release was augmented by trituration and/or mild shaking with a vortex. Additional trituration and/or mild shaking was performed after adding 2 mL of fetal bovine serum (FBS; Atlanta Biologicals, Flower Branch, Georgia, USA) to maximize the number of isolated crypts. After tissue fragments settled to the bottom of the tube, the crypt suspension containing the supernatant was transferred to a new conical tube and then centrifuged at 150g, 4 °C for 10 min. After centrifugation, the supernatant was removed, and the cell pellet was washed with 10 mL of incomplete media without an intestinal stem cell (ISC) growth factor (IMGF-) medium (Table 1) by repeating the centrifugation and decantation of the supernatant. Following crypt pellet washing, the cell pellet was re-suspended in 2 mL IMGF- medium, and the approximate number of crypt units isolated was enumerated using a hemocytometer [30].

An estimated 100 crypts were seeded per well in 30 μL of Matrigel (Corning^®^ Matrigel^®^ Growth Factor Reduced [GFR] Basement Membrane Matrix) into a 24-well plate format and incubated at 37 °C for 10 min [37]. Each well received 0.5 mL of complete media with ISC growth factors (CMGF+) supplemented with the rho-associated kinase ROCK-I inhibitor Y-27632 (StemGent) and glycogen synthase kinase 3β inhibitor CHIR99021 (StemGent) (Table 1) and was then placed in a tissue culture incubator. The CMGF + medium with ROCK-I inhibitor and glycogen synthase kinase 3β inhibitor was used for the first two days of ISC culture to enhance ISC survival and prevent apoptosis. After two days, the enteroid and colonoid cultures were replenished with CMGF+ medium without the ROCK-I inhibitor and glycogen synthase kinase 3β inhibitor every two days for 6–7 days. These freshly isolated 3D enteroid and colonoid cultures were termed Passage-0 (P0).

For subsequent passaging, enteroids and colonoids in Matrigel media were mechanically disrupted using a 1 mL syringe with a 23 g ¾” needle (trituration), collected into a 15 mL conical tube, and washed with CMGF- medium, then pelleted by centrifugation at 4 °C 100× *g* for 10 min. The pelleted enteroids and colonoids were cultured as previously described and designated as P1. This process was repeated for additional cell culture passages.

### 2.3. LPS Stimulation of Canine Enteroids and Colonoids

After four passages, colonoids from a dog with IBD and enteroids and colonoids from a dog diagnosed with an intestinal mast cell tumor were grown on 18 wells of a 24-well culture plate. Among the 18 wells, nine wells served as control cultures while nine wells served as LPS-treated cultures for these experiments. The LPS from *Escherichia coli* 026:B6 in an aqueous solution obtained from Thermo Fisher Scientific (Waltham, USA) was used for the study. To investigate the influence of a tumor microenvironment on the intestinal epithelium, we utilized enteroids from a canine intestinal mast cell tumor. In the control enteroid/colonoid cultures, the culture wells were treated with only the CMGF+ medium, whereas the LPS treatment group received the CMGF+ medium added with LPS at a concentration of 5 µg/mL. A high dose of LPS (5 µg/mL) was used to mimic the high concentration of bacterial-derived luminal LPS [38]. As small molecules are absorbed rapidly, LPS reaches the apical and basolateral sides of the organoids. After 48 h [39], photographs of the culture wells containing the organoids were obtained using a phase-contrast microscope, and the culture media were collected and stored. All enteroids and colonoids from the separate nine well groups were pooled and washed with PBS and then homogenized in TRIzol (Invitrogen™:TRIzol™ Reagent #15596026) for the microarray and quantitative real-time PCR (qPCR) analyses. Because previous work by Yin et al. (2019) [40] already provided intestinal transcriptome studies on exposure to LPS, we did not include this component in our current investigation. Instead, we utilized IBD enteroids, colonoids, and tumor enteroids as controls for the organoid groups treated with LPS. These samples were labeled separately (1–6) to avoid bias in interpretation and are described in Table 2.

### 2.4. RNA Extraction and Affymetrix Canine Genome 2.0 Array Microarray Processing

Total RNA including microRNA was isolated from these six samples using the miRNeasy Mini Kit (QIAGEN #217004) following the manufacturers’ instructions. Extracted RNA was quantitated, and the purity was evaluated using a Synergy H1 Hybrid Multi-Mode Reader (BioTek). The high-quality RNA samples were then analyzed by the Gene Expression and Genotyping Facility of the Case Comprehensive Cancer Center in Cleveland, OH. The samples were further purified using Qiagen miniprep RNA clean up (Qiagen) as per the Affymetrix protocols for array preparation. Purified RNA samples were quantified and assessed for quality using a Bioanalyzer 2100 instrument (Agilent Technologies, Palo Alto, CA, USA). A total of 0.1 µg of purified RNA was reverse transcribed and used to make cRNA. According to the manufacturer’s instructions, samples were hybridized to Canine Genome v.2.0 Affymetrix oligonucleotide arrays (Affymetrix, Santa Clara, CA, USA).

### 2.5. Assessment of Proliferation

As per the manufacturer’s instructions, samples containing 0.5 µg of total RNA were used to synthesize cDNA using the iScript™ Advanced cDNA Synthesis Kit (Bio-Rad, Life Science, Hercules, CA, USA). Real-time PCR for Ki-67 expression was carried out using the synthesized cDNA using PowerUp SYBR Green Master Mix following the manufacturer’s protocol. Thermocycling conditions were as follows: 50 °C for 2 min and then 95 °C for 2 min, followed by 35 cycles of 95 °C for 15 s and 60 °C for 1 min. The expression of *Ki-67* (sense primer: CTCCCAGGTGCTATGTTCTATC; antisense primer: ACCTACTGGCCTGAGTAAGA; Amplicon size: 100 bp) in the cell cultures was normalized using *GAPDH* (sense primer: GATGCTGGTGCTGAGTATGT; antisense primer: GTCACCCTTCTCCACCTTTATG; Amplicon size: 102 bp) and quantified using the delta-delta Ct method [41]. The surface area and [42] diameter of the organoids as well as the crypt length and the number of crypts per organoid were measured to determine the growth rate and proliferation of organoids under LPS stimulation. The organoid surface area, diameter, and crypt length as well as the total number of crypts present in an organoid were measured using Image J software [42]. An unpaired two-tailed Student’s *t*-test was used for statistical analysis.

### 2.6. Analysis of Microarray Data

Microarray CEL files were processed using the R library affylmGUI. Contrasts were computed as given below:Sample 2 vs. Sample 1Sample 4 vs. Sample 3Sample 6 vs. Sample 5Sample 5 vs. Sample 3Sample 3 vs. Sample 1Sample 6 vs. Sample 4Sample 4 vs. Sample 2

AffylmGUI [43] was used to analyze the microarray data. The quality of the microarray chips was determined by NUSE and RNA degradation plots. Probe level data were converted into probe set expression data and normalized using GCRMA. Contrasts between the treatment and control were made using the built-in linear model of affylmGUI. Since pooled data were used without replication, there were no replicates, and empirical Bayes statistics could not be computed. MA plots were generated in R, where M is the log_2_ ratio of the probe intensity, and A is the average probe intensity in the contrast (Appendix A; Probe IDs and their associated gene information are available at https://www.ebi.ac.uk/arrayexpress/files/A-AFFY-149/A-AFFY-149.adf.txt, accessed on 5 May 2020). The MA plot shows a cone shape where log_2_ fold change M decreases with the average probe intensity A. Probes that were on the edge of this cone shape are of potential interest as they fall outside the bulk of the probes for a given intensity and are the most differentially expressed (DEG) given their average probe intensity. Therefore, to identify probes of interest, the hexbin library in R was used to identify probes that fell into low-density regions of the MA plot, which corresponded to the edges of the MA plot [44]. This method was chosen to select probes without bias and for reproducibility. One advantage of this approach is that the probes that are the most different in a comparison of samples will be identified. However, one point of caution is that regardless of the M value magnitude for all sample comparisons, there will still be a relatively small number of probes identified (~500). Given the lack of the replication of samples, we realize the limitations of these results and that follow-up experiments will be required to confirm these results. KEGG ids from these probes were explored using the GenomeNet KEGG pathway search [45,46,47].

### 2.7. BLAST Search and KEGG Pathway Analyses

Blast2GO was used to determine the function and localization of differentially expressed genes (DEGs) [48]. It is a widely used annotation platform that uses homology searches to associate sequence with Gene Ontology (GO) terms and other functional annotations. Blast2GO generated Gene Ontology annotations were determined for the three sub-trees of GO, (a) biological process, (b) molecular function, (c) cellular component. The KEGG (Kyoto Encyclopedia of Genes and Genomes) [45,46,47,49,50] pathway analyses were performed by Blast2GO.

## 3. Results

### 3.1. LPS Stimulates Higher Intestinal Epithelial Cell Proliferation

LPS at a concentration of 5 µg caused an increase in the size and number of canine enteroids and colonoids, which were apparent under phase-contrast microscopy (Figure 1a). Although all enteroids and colonoids showed increased proliferation following LPS stimulation, the magnitude of proliferation was higher in tumor enteroids, followed by IBD enteroids and IBD colonoids. To confirm the magnitude of increased proliferation of enteroids and colonoids after LPS stimulation, qPCR was used to evaluate the mRNA expression of Ki-67, a nuclear proliferation marker [51]. LPS stimulation caused an ~4-fold increase in the Ki-67 mRNA expression in both tumor enteroids and IBD enteroids, whereas LPS caused an ~1.5-fold increase in the IBD colonoids (Figure 1b). Genes associated with proliferation such as proliferating cell nuclear antigen (*PCNA*), prominin 1 (*PROM1*), HOP homeobox (*HOPX*), and olfactomedin 4 (*OLFM4*) were also upregulated in LPS treated IBD intestinal organoids and tumor enteroids (Figure 1c). Increased surface area and diameter of the organoids as well as the crypt length and the number of crypts per organoid were all detected after LPS treatment, indicating that LPS stimulation increased the growth rate of organoids (Figure 1d).

### 3.2. Microarray Analysis Reveals Differential Expression of Genes Stimulated by LPS in IBD Intestinal Organoids and Tumor Enteroids

In this study, we used the GeneChip^®^ Canine Genome 2.0 Array that contains more than 42,800 *Canis familiaris* probe sets to evaluate gene expression [52]. These probe sets targeted more than 18,000 *C. familiaris* mRNA/EST-based transcripts and more than 20,000 non-redundant predicted genes for more comprehensive coverage [52]. We performed multiple comparisons between samples to determine the effect of LPS treatment, anatomical location, and the in vivo microenvironment (tumor/IBD) on the mRNA expression profiles of organoids following LPS stimulation. The highest number of highly differentially expressed mRNAs (684) was found in LPS-treated IBD colonoids vs. the control IBD colonoids (Table 3). Among this total number of highly differentially expressed mRNAs, 48% were upregulated, and 52% were downregulated (Table 3). Although similar numbers of differentially expressed mRNAs (677) were found between the LPS-treated tumor enteroids vs. the control tumor enteroids, 56% were upregulated, and 44% were downregulated.

The next highest number of differentially expressed mRNAs (639) was found between the LPS treated IBD enteroids vs. control untreated IBD enteroids (Table 3). Of the differentially expressed mRNAs, 49% were upregulated, and 51% were downregulated (Table 3). Interestingly, between the LPS treated IBD colonoids and enteroids, only 421 differentially expressed mRNAs were identified, of which 39% were upregulated and 61% were downregulated. LPS stimulation caused 314 differentially expressed mRNAs in the IBD enteroids vs. the LPS treated tumor enteroids. Among these differentially expressed mRNAs, 51% were upregulated, and 49% were downregulated. Furthermore, LPS stimulation resulted in a unique transcriptomic heat map for tumor enteroids, IBD enteroids, and colonoids (Appendix A).

### 3.3. Gene Expression Profiles and Pathway Enrichment Analysis

#### 3.3.1. The LPS Treated Tumor Enteroids vs. the Control Tumor Enteroids

Genes that demonstrated high levels of activity in tumor enteroids following LPS stimulation included KH domain-containing, RNA-binding, signal transduction-associated protein 3 (*KHDRBS3*), Flap endonuclease GEN homolog 1 (*GEN1*), krev interaction trapped-1 (*KRIT1*), Centromere protein F (*CENPF*), ciliogenesis and planar polarity effector complex subunit 1 (*CPLANE1*), and tyrosine-protein kinase (*STYK1*). However, expressions of quinone oxidoreductase-like protein 2/Crystallin Zeta Like 2 (*LOC610994*), G patch domain-containing protein 4 (*Gpatch4*), Solute Carrier Family 7 Member 1/high-affinity cationic amino acid transporter 1 (*SLC7A1*), cation-transporting ATPase 13A2 (*ATP13A2*), and testis expressed 45 (*TEX45*) genes were significantly repressed in LPS-treated tumor-adjacent enteroids (Figure 2 and Table 4).

Using the Blast2GO functional group analysis [48,50] (Figure 2; Table 4), the highly differentially expressed genes (DEGs) stimulated by LPS treatment on tumor enteroids were found predominantly overrepresented in biological processes involved in cellular metabolic process and regulation, organic substance metabolic process/nitrogen compound metabolic process (18%), regulation of molecular function, cellular homeostasis/regulation of biological quality (7%), and cell cycle checkpoint/microtubule-based process (4%). In comparison, categories corresponding to identical protein binding (14%) and nucleic acid binding (14%) were significantly overrepresented in molecular function. However, in the cellular component category, the DEGs were overrepresented for intracellular organelles (17%), intracellular non-membrane-bounded organelles (10%), membrane-bounded organelles (10%), an integral component of membrane (10%), non-membrane-bounded organelles (10%), plasma membrane (7%), and centrosome (7%) (Figure 2; Table 4).

#### 3.3.2. The LPS Treated IBD Enteroids vs. the Control IBD Enteroids

Genes that demonstrated high levels of activity in IBD enteroids following LPS stimulation included the DENN domain containing 5B (*DENND5B*), serine/threonine/tyrosine-interacting-like protein 1 (*STYXL1*), regenerating islet-derived protein 3-gamma-like (*REG3G*), and protein S100-A16 (*S100A16*). However, expressions of protein FAM122B (*FAM122B*), centrosomal protein of 70 kda (*CEP70*), beta-1,4-mannosyl-glycoprotein 4-beta-N-acetylglucosaminyltransferase (*MGAT3*), and mucin 1 (*MUC1*) genes were significantly repressed in the LPS-treated IBD enteroids (Figure 3).

The DEGs stimulated by LPS treatment on the IBD enteroids were found to be predominantly overrepresented in the biological process category for the organonitrogen compound metabolic process, microtubule cytoskeleton organization, regulation of microtubule-based process, cellular component biogenesis/organization and regulation, and the phosphorus/cellular macromolecule/protein metabolic process using Blast2GO functional group analysis [48,50] (Table 5). In comparison, the categories corresponding to phosphoprotein phosphatase activity, identical protein binding, cytoskeletal protein binding, hydrolase activity, acting on ester bonds, and cation binding were significantly overrepresented in molecular function. However, in the cellular component category, the DEGs were overrepresented for intracellular non-membrane-bounded organelles (14%) and the membrane/intrinsic component of membrane, cytoplasm (14%) (Table 5).

#### 3.3.3. The LPS Treated IBD Colonoids vs. the Control IBD Colonoids

Genes such as collagenase 3 (*MMP13*), dual oxidase maturation factor 2 (*DUOXA2*), ceruloplasmin (*CP*), UMP-CMP kinase 2, mitochondrial (*CMPK2*), thrombospondin-4 (*THBS4*), and interleukin-8 (*CXCL8*) showed elevated levels of activity in the IBD colonoids after LPS treatment while the colonoids exposed to LPS had decreased expressions of abl interactor 2 (*ABI2*), ubiquitin-conjugating enzyme E2 E3 (*UBE2E3*), echinoderm microtubule-associated protein-like 2 (*EML2*), extracellular sulfatase Sulf-2-like (*SULF2*), and ectonucleotide pyrophosphatase/phosphodiesterase family member 6 (*ENPP6*) genes (Figure 4).

Using the Blast2GO functional group analysis, it was noticed that the DEGs stimulated by LPS treatment on the IBD cololoids were predominantly overrepresented in the biological process category for the primary metabolic process, organic substance/nitrogen compound metabolic process (14%), establishment of localization (9%), regulation of biological quality, cellular homeostasis, oxidation–reduction process (5%), and cell communication, signal transduction, response to chemical/external stimulus, cellular response to stimulus, immune response (5%) (Table 6). When it comes to molecular function, the categories that correspond to metal ion binding (19%), cytokine receptor binding (6%), and oxidoreductase activity, oxidizing metal ions, oxygen as acceptors (6%) were significantly overrepresented. However, in the category of cellular components, the DEGs were overrepresented in the extracellular region (28%) and membrane (17%) (Table 6).

#### 3.3.4. The Control IBD Colonoids vs. the Control IBD Enteroids

Carbonic anhydrase 1 (*CA1*), protein S100-A16 (*S100A16*), mucin-1 (*MUC1*), and KH domain-containing, RNA-binding, signal transduction-associated protein 3 (*KHDRBS3*) showed increased activity in the IBD colonoids compared to the IBD enteroids, while bile acid-CoA: amino acid N-acyltransferase-like (*BAAT*), annexin A13 (*ANXA13*), and aminopeptidase N (*ANPEP*) exhibited decreased expression levels (Figure 5).

With the help of the Blast2GO functional group analysis, it was observed that the top upregulated or downregulated DEGs in the IBD colonoids, compared to the IBD enteroids, were predominantly overrepresented in the biological process category, specifically in the cellular metabolic process (7%) and response to stress, regulation of metabolic process/molecular function/hydrolase activity (2%) (Table 7). In molecular function, the categories that correspond to metal ion binding (21%) and RNA binding (8%) were significantly overrepresented while in the cellular component categories, the DEGs were overrepresented in the membrane, cell membrane (14%), and cytoplasm and cytosol (12%) categories (Table 7).

#### 3.3.5. The Control IBD Enteroids vs. the Control Tumor Enteroids

The activity of CA1, protein S100A16 (*S100A16*), beta-site APP-cleaving enzyme 2 (memapsin 1) (beta-secretase 2) (*BACE2*), and mucin-1 (*MUC1*) was increased in the IBD enteroids compared to the tumor enteroids, while the activity of bile acid-CoA:amino acid N acyltransferase-like (*BAAT*), annexin A13 (*ANXA13*), and aminopeptidase N (*ANPEP*) was decreased (Figure 6).

With the help of the Blast2GO functional group analysis, we observed that the top upregulated or downregulated DEGs in the IBD enteroids, compared to tumor enteroids, were predominantly overrepresented in the biological process category, specifically in the primary metabolic process, organic substance metabolic process (10%), and cell communication, signal transduction, cellular response to stimulus (4%) (Table 8). In the molecular function, the categories that corresponded to ion binding (18%), hydrolase activity (14%), and enzyme regulator activity (11%) were significantly overrepresented, while in the cellular component categories, the DEGs were overrepresented in the cytoplasm (13%), membrane/plasma membrane (13%), organelle (11%), and extracellular region (10%) categories (Table 8).

#### 3.3.6. The LPS Treated IBD Colonoids vs. the LPS Treated IBD Enteroids

The activities of *CA1*, *S100A16*, annexin A10 (*ANXA10*), and *MUC1* were increased in the LPS treated IBD colonoids compared to the LPS treated tumor enteroids, while the activities of *BAAT*, long-chain-fatty-acid-CoA ligase 5 (*ACSL5*), ectonucleotide pyrophosphatase/phosphodiesterase family member 6 (*ENPP6*) were decreased (Figure 7).

With the help of the Blast2GO functional group analysis, it was noticed that the top upregulated or downregulated DEGs in the LPS treated IBD colonoids compared to the LPS treated IBD enteroids were predominantly overrepresented in the biological process category, specifically in the cellular metabolic process (7%) and signal transduction, cell communication, cellular response to stimulus (3%) categories (Table 9). In molecular function, the categories that corresponded to cation binding (17%), hydrolase activity, acting on ester bonds (10%), and signaling receptor binding (7%) were significantly overrepresented, while in the cellular component categories, the DEGs were overrepresented in the intracellular organelle (14%) and membrane-bounded organelle (14%) categories (Table 9).

#### 3.3.7. The LPS Treated IBD Enteroids vs. the LPS Treated Tumor Enteroids

The activities of *CA1*, *ANXA10*, and *MUC1* were increased in LPS treated IBD enteroids compared to the LPS treated tumor enteroids, while the activities of *BAAT*, *ACSL5*, and *ENPP6* were decreased (Figure 8).

With the help of the Blast2GO functional group analysis, it was noticed that the top upregulated or downregulated DEGs in the LPS treated IBD enteroids compared to the LPS treated IBD tumor enteroids were predominantly overrepresented in the biological process category, specifically in the regulation of the cellular process (7%), cellular metabolic process (6%), regulation of molecular function (5%), and cell communication, signal transduction, cellular response to stimulus (4%) categories (Table 10). In the molecular function, the categories that corresponded to the receptor ligand activity (14%), metal ion binding (14%) and cytokine receptor binding, G protein-coupled receptor binding (9%) were significantly overrepresented, while it in the cellular component categories, the DEGs were overrepresented in the plasma membrane (23%), intracellular organelle, membrane-bounded organelle (15%), and integral component of the membrane (15%) (Table 10).

### 3.4. Genes Showing Similar Expression Patterns between IBD Enteroids and Colonoids Following LPS Stimulation

The pattern of gene expression was compared between the LPS-stimulated IBD enteroids and colonoids. We identified 25 genes across these two groups with similar expression patterns (Figure 9 and Table 11 and Table 12). Following LPS stimulation, 14 of these 25 genes were elevated in both the IBD enteroids and colonoids, whereas 11 were downregulated. These 25 genes may represent the intestinal epithelium signature LPS responsive gene network, regardless of the intestinal regions.

Following LPS stimulation of the IBD enteroids and colonoids, significant upregulation of the genes involved in stress response (*OAS1*, *OASL*, *IFIT1*, *GPX1*, and *ISG15*) was observed, even though several of these genes (*OAS1*, *OASL*, and *ISG15*) and *EEF1A1* were previously implicated in primary metabolic processes and their regulation [53,54,55,56]. Additional upregulated genes include those involved in the cellular response to stimulus (*TFF1*, *IFIT1*, *ISG15*, *GPX1*, *IGFBP1*), signal transduction (*TFF1*, *IGFBP1*, *ISG15*), response to biotic stimulus, immune effector process, immune system regulation, immune system development/cytokine production (*OAS1*, *OASL*, *IFIT1*, *ISG15*), antigen processing and presentation (*TAP2*, *DLA class I*), oxidation–reduction process and cellular detoxification (*CP*, *GPX1*), regulation of molecular function (*OASL*, *IFIT1*, *NOXO1*), transmembrane transport (*TAP2*), and positive regulation of the viral process (*IFIT1*) (Figure 9 and Table 11). On the other hand, considerable downregulation of the *RPS24* and *ATP1A1* genes, involved in primary metabolic processes and transmembrane transport, respectively, was observed (Figure 9 and Table 11).

### 3.5. Genes Upregulated in IBD Enteroids and Downregulated in IBD Colonoids (and Vice Versa) Following LPS Stimulation

Interestingly, 20 genes were found to be altered in the opposite direction (Figure 10 and Table 13 and Table 14). These 18 genes were elevated in the LPS-treated IBD enteroids; however, these same 20 genes were downregulated in the LPS-treated colonoids. These findings imply that a distinct gene signature might be utilized to differentiate the intestinal region in response to LPS stimulation. 

The LPS-stimulated genes that exhibited the opposite expression trends between the IBD enteroids and colonoids were involved in anion transport (*TOMM20*, *ANXA1*, *KPNA2*), translation (*RPS20*, *EIF3F*, *MRPS21*), protein import (*TOMM20*, *KPNA2*), protein-containing complex assembly (*TOMM20*, *SF3A1*), regulation of proteolysis (*LOC111092171*, *HSPD1*), RNA metabolic process (*GTF2A2*, *SF3A1*), positive regulation of leukocyte cell–cell adhesion, T-cell activation, apoptotic process, and the regulation of adaptive immune response (*ANXA1*, *HSPD1*), as shown by the Blast2GO analyses [48,50] (Figure 10 and Table 13 and Table 14). In the IBD colonoids, LPS treatment increased the expression of the *EIF3F*, *ANXA1*, *SF3A1*, and *LOC111092171* genes, whereas it decreased their expression in the IBD enteroids (Figure 10). In contrast, the expression of *RPS20*, *MRPS21*, *TOMM20*, *KPNA2*, *HSPD1*, and *GTF2A2* genes was elevated and reduced in the LPS treated IBD enteroids and colonoids, respectively (Figure 10). These unique gene expression signatures in the IBD enteroids and colonoids treated with LPS may provide critical insights into the intestinal physiology and novel inflammatory pathways that may serve as future drug discovery targets.

### 3.6. KEGG Pathway Analyses

The KEGG [45,46,47,49] pathway analysis (https://www.kegg.jp/kegg/kegg1.html, accessed on 5 May 2020) displayed that the most up- or downregulated DEGs in the IBD intestinal organoids and tumor enteroids were involved in metabolic pathways related to nitrogen metabolism (*CA1*), glutathione metabolism, renin–angiotensin system, hematopoietic cell lineage (*ANPEP*/*LOC112653425*), primary bile acid biosynthesis, taurine and hypotaurine metabolism, biosynthesis of unsaturated fatty acids, peroxisome, and bile secretion (*BAAT*). *BACE2*, which is involved in Alzheimer’s disease, was found to be one of the most prevalent DEGs in the IBD enteroids compared to the tumor enteroids. However, under LPS stimulation, in addition to *CA1* and *BAAT*, a gene involved in fatty acid biosynthesis and degradation, the PPAR signaling pathway, peroxisome, ferroptosis, thermogenesis, and adipocytokine signaling pathway (*ACSL5*) as well as the *ENPP6* gene involved in ether lipid metabolism, were among the top up- or downregulated DEGs in the IBD intestinal organoids and tumor enteroids (Figure 11 and Table 15).

As a result of LPS stimulation, genes that were expressed similarly in both the IBD enteroids and colonoids were shown to be involved in the metabolism of porphyrin and ferroptosis (*CP*), ABC transporters, phagosome, antigen processing and presentation, human cytomegalovirus infection, Herpes simplex virus 1 infection, Epstein–Barr virus infection, human immunodeficiency virus 1 infection, primary immunodeficiency (*TAP2*/*ABCB3*), arachidonic acid, glutathione metabolism (*GPX1*), and nucleocytoplasmic transport, legionellosis, and leishmaniasis (*EEF1A1*) (Table 15).

## 4. Discussion

In this study, we generated enteroids and colonoids from dogs diagnosed with IBD or intestinal mast cell tumor. The goal of these experiments was to better understand how intestinal epithelial cells respond to LPS stimulation under different pathological conditions. We further investigated the differences between the organoids derived from various intestinal compartments (small intestine vs. colon) in IBD dogs following LPS stimulation. IBD patients have long-lasting epithelial abnormalities that may be acquired during disease development, which contributes to the progression of the pathology. Genetic mutations associated with human IBD are involved in colonic epithelium architectural organization, resulting in the establishment of a permanent epithelial phenotype in organoid cultures [57]. However, recently published results on inflammatory bowel disease (IBD) indicate that patient-derived organoids may lose their inflammatory signature after 5–6 passages [58,59], and hence we decided to examine an earlier time point. However, it has been reported that for at least 17 passages, intestinal organoids express most gene orthologs that are particularly expressed in the human gut, suggesting that adult stem cells preserve intestine-associated gene expression across long-term culture conditions [60,61,62]. Transcriptomic analyses using microarrays [63] were used to identify differentially expressed genes in enteroids and colonoids derived from diseased dogs. These data were the first to characterize regional specific transcriptomic changes in the intestinal epithelial cells of dogs with IBD and intestinal cancer in response to ex vivo LPS stimulation.

An increase in proliferation in tumor enteroids and both IBD enteroids and colonoids was observed following LPS stimulation in the present study, as indicated by Ki-67 estimates, stem cell markers as well as an increase in the surface area and diameter of the organoids, and an increase in crypt length and the number of crypts per organoid. Adult stem cells expressing the leucine-rich repeat-containing G-protein coupled receptor (LGR5) could be identified close to the base of the intestinal crypts [64] that were employed for organoid proliferation. Using canine-specific LGR5 probes, we previously detected ISCs within the intestine organoid crypts [30], demonstrating that proliferation is restricted to crypt-like budding structures [64]. Ki67 has been used as a proliferation marker as its expression is strongly associated with proliferation and growth [65,66]. Other intestinal stem cell markers and genes associated with proliferation, notably proliferating cell nuclear antigen (*PCNA*), prominin 1 (*PROM1*), HOP homeobox (*HOPX*), and olfactomedin 4 (*OLFM4*), were also increased in LPS-treated IBD intestinal organoids and tumor enteroids. *PCNA* is widely accepted as a proliferation marker [67], while PROM1 [68] is a marker for stem cells and early progenitors in the mouse small intestine. Likewise, *OLFM4* is a highly specific marker reported in murine and human intestinal stem cells and its expression is regulated by the *NF-κB*, Notch, and Wnt signaling pathways in digestive diseases [69]. The role of OLFM4 in pathogenesis relates to anti-inflammation, apoptosis, cell adhesion, and also proliferation, as it regulates the cell cycle and promotes the S phase transition [69,70]. While *HOPX*-expressing cells give rise to crypt base columnar (CBC) cells and all mature intestinal epithelial lineages, CBCs can give rise to +4 *HOPX* positive cells. It is also possible that LPS-induced enhanced proliferation in tumor enteroids was caused by the increased expression of genes involved in the cell cycle process such as centromere protein F (*CENPF*) and flap endonuclease GEN homolog 1 (*GEN1*) as well as a receptor protein tyrosine kinase (*STYK1*). *CENP-F* is a centromere-kinetochore complex-associated protein, and its expression is increased in tumors [71] and it serves as a marker for cell proliferation in human malignancies [72]. CRISPR-Cas9 silencing of *CENPF* in human prostate cancer cells resulted in decreased cell proliferation [73,74]. *CENPF* regulates cancer metabolism by modulating the phosphorylation signaling of pyruvate kinase M2 [74]. Similarly, *GEN1* expression was induced in tumor enteroids. GEN1, similar to other members of the Rad2/XPG family as *FEN1*, is a monomeric 59-flap endonuclease, but it can dimerize on Holliday junctions (HJs), providing the two symmetrically aligned active sites required for HJ resolution [75]. Members of the Rad2/XPG family such as *FEN1* are significantly expressed in proliferating cell populations, consistent with their role in DNA replication [76]. For instance, its expression is associated with the proliferation of mammary epithelial cells [76]. *FEN1* expression is increased in metastatic prostate cancer cells, gastric cancer cells, pancreatic cancer cells, and lung cancer cell lines, and with tumor progression [77]. Additionally, the receptor STYK1 is required for cell proliferation [78]. Oncogenic potential of STYK1 has been studied widely in gallbladder cancer (GBC) and was reported to be largely dependent on the PI3K/AKT pathway. The tumor-stimulating activity of STYK1 was abolished by the AKT-specific inhibitor MK2206 as well as by STYK1 gene silencing [78]. A crucial NF-κB regulator, Sam68 (KHDRBS3), was also observed to be elevated in LPS-treated tumor enteroids, along with the genes *GEN1*, *KRIT1*, *CENPF*, and *STYK1*. Sam68 (KHDRBS3) exhibited a prognostic significance in various malignancies and was elevated in cancer cell lines [79,80,81]. Sam68 (KHDRBS1), a homolog of KHDRBS3, was found to be co-expressed with cancer-related genes in the genome-wide analysis [79,80]. Our results suggest the possible roles of *GEN1*, *KRIT1*, *CENPF*, *STYK1*, and *Sam68/KHDRBS3* in promoting cancer cell proliferation, and these findings may provide a potential therapeutic target to control mast cell malignancy.

Even though LPS treatment failed to diminish the expression of genes involved in proliferation and malignancy such as *GEN1*, *KRIT1*, *CENPF*, *STYK1*, and *Sam68*/*KHDRBS3*, it did reduce the expression of numerous genes implicated in tumor growth. LPS treatment, for example, reduced the expression of the *LOC610994* (zeta crystallin), *Gpatch4*, *SLC7A1*, *ATP13A2*, and *TEX45* genes. Both *TEX45* [82] and zeta crystallin (*CryZ*) [83] are expressed in several human cancer types including colorectal cancer. While *CryZ* may protect cancer cells from oxidative stress and help them maintain an optimal DNA complement, it can fuel the cancer requirement for energy by activating glutaminolysis. Furthermore, increased expression and/or activation of *CryZ* following acidification of the tumor microenvironment may contribute to cancer cell survival and drug resistance by increasing the anti-apoptotic gene expression in cancer cells [83]. *GPATCH4* was identified in melanoma patient sera and was revealed to be increased in hepatocellular carcinoma [84]. Similarly, the SLC7A1/CAT1 arginine transporter plays a critical function in colorectal cancer by increasing arginine metabolism [85]. After LPS treatment, another important gene, *ATP13A2*, was downregulated in tumor enteroids. *ATP13A2* regulates autophagy, as demonstrated by *ATP13A2* knockdown, decreasing cellular autophagy levels, reversing ATP13A2-induced stemness in colon cancer cells with the autophagy inhibitor bafilomycin A1 [86], and the reduction in the volume of colon cancer xenografts in mice treated with *ATP13A2* siRNA [86]. While all of these pieces of evidence point to *ATP13A2* and autophagy, studies have also revealed a link between the level of *ATP13A2* expression and the survival rate of colon cancer patients. Colon cancer patients with elevated ATP13A2 expression displayed shorter overall survival than those with low ATP13A2 [86].

Interestingly, while LPS treatment increased the proliferation and expression of the *GEN1*, *KRIT1*, *CENPF*, *CPLANE1*, *STYK1*, and *KHDRBS3* genes, it decreased the expression of the cancer associated *LOC610994*, *Gpatch4*, *SLC7A1*, *ATP13A2*, and *TEX45* genes [75,81,82,83,87] in tumor enteroids. This could imply that LPS has some anticancer activity. LPS treatment has been shown to have potent anticancer effects [88,89] against adoptively transferred tumors (TA3/Ha murine mammary carcinoma) in mice [90,91]. LPS and the other two potent TLR4 agonists have been proven to be effective in treating a variety of carcinomas [89,92]. In addition, research on mice with lung tumors has demonstrated that a modest dose of LPS promotes tumor development, whereas a large amount induces tumor regression [90]. Plasmacytoid dendritic cells are required for LPS to exert its dose-dependent effects [90].

The addition of LPS increased the production of proteins involved in cytokine signaling, most notably interferon and interleukin signaling [93]. After LPS treatment, we observed the upregulation of genes involved in biotic stimuli and immune effector processes (*OAS1*, *OASL*, *IFIT1*, *ISG15*) as well as antigen processing and presentation (TAP2, DLA class I) and signal transduction (*TFF1*, *IGFBP1*, *ISG15*). Emerging evidence suggests that the increased activity of these genes contributes to the inflammatory response induced by LPS. For example, LPS and interferons highly activate the ubiquitin-like protein ISG15 and 2′-5′-oligoadenylate synthetase-like (OASL) [56,94]. OASL is necessary for the antiviral signaling pathway mediated by IFNs [56] as it regulates pro-inflammatory mediators such as cytokines and chemokines [56]. Likewise, OAS1 has a strong antiviral impact both in vivo and in vitro [95] and protects cells from viral infection [96]. When exogenous recombinant OAS1 was added to the cultured cells, it was internalized and exerted a potent antiviral effect [95]. LPS has also been shown to induce OAS in marine sponges [55]. TAP2 and trefoil factor 1 (TFF1) upregulation in IBD enteroids and colonoids following LPS treatment could result from their ability to modulate TLR4 signaling or their anti-inflammatory properties [97,98,99]. TAP2 has been shown to inhibit TLR4 signaling and diminish the systemic cytokine response induced by LPS [97]. TFF1 exhibits anti-inflammatory properties, and its involvement as a tumor suppressor gene has been shown through studies using the Tff1-knockout (Tff1-KO) mouse model [100,101]. While the Tff1-KO mice developed gastric malignancies via the activation of NF-κB and chronic inflammatory pathways [98], treatment with exogenous TFFs alleviated the gastrointestinal inflammation [101,102]. Insulin-like growth factor-binding protein 1 (IGFBP1) is a member of the GFBPs family of proteins that interact with insulin-like growth factors (IGFs) to regulate their anabolic activity [103]. While LPS injection reduces the circulating IGF-I levels, it increases the IGFBP-1 levels [104,105]. LPS has also been shown to induce IGFBP-1 in the liver, muscle, and kidney tissues [106]. Similarly, NADPH oxidase 1 (NOX1) activity induced by LPS was detected in dendritic cells [107], corroborating our current findings. NOX1 requires two additional proteins, NOXO1 and NOXA1, and it interacts with p22phox in a complex [108]. NOX1 is triggered by the GTPase Rac, which binds directly to it or via the NOXA1 TPR domain [108]. Genes such as *CP* and *GPX1*, involved in the oxidation–reduction process and cellular detoxification, were expressed more abundantly in the IBD intestine organoids following LPS treatment. The generation of reactive oxygen species (ROS) during the innate immune response to LPS is one of the anti-pathogen responses that can cause oxidative damage, and GPx-1, an antioxidant enzyme [109,110], can protect the intestine from such damage. Gpx-1 facilitates the generation of pro-inflammatory cytokines in response to LPS exposure [111]. GPx1 has been shown to regulate LPS-induced adhesion molecule expression in endothelial cells through modulating CD14 expression. Suppression of GPx-1 promotes the expression of the CD14 gene in human microvascular endothelial cells [112]. GPx-1 deficiency increases LPS-induced intracellular ROS and CD14 and intercellular adhesion molecule-1 (ICAM-1) expression [112]. Studies have shown a connection between LPS stimulation and NO production and the Ceruloplasmin (CP) activity [113]. Without changing the iNOS expression levels, CP enhances the LPS-activated iNOS activity. An unknown Cp receptor activates this intracellular signaling that cross-talks with the response stimulated by LPS [114]. Members of the S100 family, S100P and S100A16, were also induced by LPS stimulation and IBD; a similar induction of S100 protein members and their role in the regulation of inflammation have been reported in previous studies [115,116]. Furthermore, our current research shows the associations between the S100 proteins and IBD, with LPS and IBD both stimulating S100A16 activity. Our present study, along with earlier reports [117], points to the potential use of S100 markers for diagnostic purposes, specifically S100A16 in IBD. The use of S100 A proteins in canine feces as biomarkers of inflammatory activity has previously been reported [118]. The transmembrane ion pump Na+/K+-ATPase (ATP1A1) has been linked to nuclear factor kappa B (NFκB) signaling [119], a signal associated with the LPS induced immune response. While α2Na+/K+-ATPase haploinsufficiency was reported to regulate LPS-induced immune responses negatively, we observed that *ATP1A1* was suppressed in both the IBD enteroids and colonoids treated with LPS in the present study. Nonetheless, our demonstration of the inhibited expression of *ATP1A1* in IBD intestinal organoids identifies it as a promising candidate for further analysis.

The GI tract microbial ecology varies according to its microbial diversity and anatomic location. Oxygen tension is a primary determinate of microbial numbers and complexity [120]. The upper GI tract, stomach, and small intestine have a lower pH, shorter transit time, and reduced bacterial population. However, the colon harbors the greatest diversity of bacteria as it has a low cell turnover rate, a low redox potential, and a longer transit time [121]. As a result of the LPS reaction, the gene expression profile of enteroids and colonoids from the IBD dogs also altered. Twenty DEGs were identified as exhibiting an opposite expression trend (i.e., upregulated in IBD enteroids and downregulated in IBD colonoids and vice versa) following LPS stimulation. These 20 unique signature genes might be used to differentiate the intestinal regions in response to LPS stimulation.

The differential expression of *TOMM20* and *eIF3F* between the IBD enteroids and colonoids may be a reason why IBD enteroids with a higher TOMM20 expression and lower eIF3F expression proliferated more than the colonoids (as indicated by Ki-67 expression and number of crypts/organoid). Mitochondrial protein, translocase of the outer mitochondrial membrane complex subunit 20 (TOMM20), promotes the proliferation and resistance to apoptosis and serves as a marker of mitophagy activity [122]. Reduced *TOMM20* expression in response to LPS treatment implies that LPS activates mitophagy [123], resulting in decreased proliferation in the IBD colonoids (as observed in the current study). In the tumor cells, enhanced *eIF3F* expression inhibits translation, cell growth, and cell proliferation and induces apoptosis, whereas knockdown of eIF3f inhibits apoptosis, showing the role of eIF3f as an essential negative regulator of cell growth and proliferation [124,125]. Furthermore, the expression of *FAM168A(TCRP1)* in IBD enteroids implies that IBD enteroids are more protected than colonoids against LPS stimulation. FAM168A operates via the PI3K/AKT/NFKB signaling pathway and has previously been shown to protect cells against apoptosis [126].

Several genes that participated in RNA metabolism, protein synthesis, import, protein complex assembly and proteolysis, anion transport, adaptive immune response, and apoptosis were also differentially expressed in the LPS-treated IBD colonoids and enteroids. For instance, LPS treatment enhanced the expression of SF3A1, S100P, CRIP1, ANXA1, and RGS2 in the IBD colonoids. The presence of SF3A and SF3B is required for a robust innate immune response to LPS and other TLR agonists [127]. SF3A1, a member of the SF3A complex, regulates LPS-induced IL-6 by primarily inhibiting its production [127]. Similarly, S100 proteins are known to be secreted in response to TLR-4 activation. S100 proteins also influence proliferation, differentiation, and apoptosis, in addition to inflammation [116]. Earlier reports and our recent observation of enhanced cysteine-rich intestinal protein 1 (CRIP1) expression in response to LPS suggest that CRIP may play a role in immune cell activation or differentiation [128]. While CRIP1 is abundant in the intestine [129], it is abnormally expressed in certain types of tumor [128]. CRIP1 inhibits the expression of Fas and proteins involved in Fas-mediated apoptosis [128]. LPS activates the *AnxA1* gene significantly, and in the absence of *AnxA1*, LPS induces a dysregulated cellular and cytokine response with a high degree of leukocyte adhesion. The protective role of *AnxA1* was demonstrated in *AnxA1*-deficient mice. In *AnxA1*-deficient mice, LPS induced a toxic response manifested by organ injury and lethality, restored by the exogenous administration of AnxA1 [130,131]. In contrast, following LPS treatment, *GTF2A2* expression was increased in the IBD enteroids. *GTF2A2* is required for NF-kB signaling in the LPS-induced TNF responsive module [132]. Additionally, LPS promoted *HSPD1* and *Cyr61* expression in the IBD enteroids. *Cyr61* is known to be activated by LPS and may have pleiotropic responses to LPS [133]. Human hsp60 directly promotes nitrite production and cytokine synthesis in macrophages. Human hsp60 was found to synergize with IFN-γ in its proinflammatory activity [134]. The inflammatory response to LPS was evaluated in *RGS2^−/−^* mice, which exhibited a higher expression of TNF and phosphorylated p38 levels in cardiomyocytes. This study demonstrates that *RGS2* plays a function in cardioprotection and anti-inflammatory signaling via p38 [135]. *RGS2* inhibits G protein-coupled receptor signaling by increasing the rate of G protein deactivation or by decreasing the G protein–effector interactions [136].

The KEGG [45,46,47,49] pathway analysis showed the involvement of several metabolic pathway genes in the IBD intestinal organoids and tumor enteroids. For example, BAAT, which is involved in primary bile acid biosynthesis, taurine and hypotaurine metabolism, biosynthesis of unsaturated fatty acids, peroxisome, and bile secretion pathways, was identified. Bile acids (BAs) play key roles in intestinal metabolism and cell signaling and changes in BAs have also been demonstrated to influence gut homeostasis and contribute to the pathogenesis of IBD [137]. Several BA receptors have been reported to be involved in the development of colorectal cancer, cholangiocellular carcinoma, or their association with cancer cell lines [138]. The gene *ANPEP*/*LOC112653425*, involved in glutathione metabolism, the renin–angiotensin system, and hematopoietic cell lineage, was found to be one of the most prevalent DEGs in IBD enteroids and tumor enteroids. High levels of *CD13*/*ANPEP* expression have been observed in a variety of solid tumors [139]; additionally, *CD13* has been shown to be associated with malignant activity in colon and prostate cancers [140]. Soluble CD13/*ANPEP* has several pro-inflammatory functions via its interaction with G protein-coupled receptors. CD13 is significant in the etiology of numerous inflammatory disorders by regulating the growth and activity of immune-related cells as well as functions of inflammatory mediators [141]. It has been suggested that IBD inflammation may be exacerbated by a disruption of the renin–angiotensin system [142]. Additionally, inflammation contributes significantly to the etiology of Alzheimer’s disease (AD) by activating C/EBPβ/δ-secretase and causing AD-associated disorders in the gut, which are then transferred to the brain via the vagus nerve [143]. BACE1 and BACE2 β-secretases have been widely investigated in the context of Alzheimer’s disease. Specifically, BACE2 is overexpressed in tumors. BACE2 increases tumor growth by hyperactivating the NF-κB pathway via a series of phosphorylation cascades of different members of this pathway [144]. We also identified that BACE2, participating in the Alzheimer’s disease pathway, was one of the top DEGs in both the IBD and tumor enteroids in the current study.

Additionally, LPS treatment altered the *ACSL5*, *ENPP6*, and *CA1* expression in the intestine and tumor enteroids. ACSL5 is required for the de novo synthesis of lipids and fatty acid degradation, and its role in inflammation and cancer development has been reported. ACSL5 interacts with proapoptotic molecules and suppresses proliferation [145]. Following LPS treatment, genes involved in the glutathione metabolism pathway were significantly altered in serum, whereas genes involved in the bile acid biosynthesis pathway were significantly changed in numerous rat tissues [146]. LPS treatment also altered the expression of arachidonic acid, glutathione, and the nitrogen metabolic pathway genes in the IBD enteroids and colonoids. Earlier reports on the effects of LPS on nitrogen metabolism (*CA1*) in beef steers [147] and altered expression of the ABCB transporters associated with IBD [148] also corroborate the current study. The arachidonic acid (AA) pathway is implicated in a variety of inflammatory diseases [149,150]. The glutathione (GSH) pathway is a critical metabolic integrator in T-cell-mediated inflammatory responses [151]. The metabolism of AA results in the generation of reactive oxygen species (ROS) [152]. GSH and its metabolic enzymes protect tissues from oxidative damage [109,110,153,154,155,156,157,158,159,160,161]. By interacting in both the AA and GSH pathways, *GPX1* is a critical determinant of the AA effects [152]. Additionally, we identified changes in the expression of genes implicated in these metabolic pathways in the LPS-stimulated IBD intestinal organoids.

The peroxisomal metabolic gene, BAAT, was found to be among the top DEGs in both the IBD intestinal organoids and tumor enteroids. Peroxisomes have been implicated in the pathogenesis of inflammation and cancer. Peroxisomes are known to operate as immunometabolic hubs, producing and metabolizing ROS as well as unsaturated fatty acids such as docosahexaenoic acid (DHA), which can affect the inflammatory pathways [162,163]. Peroxisomal lipid production and redox balance can aid in the survival of cancer cells in the tumor microenvironment. The disruption of the interplay between peroxisomes and other cellular organelles such as the endoplasmic reticulum and the mitochondria has the potential to rewire the metabolism of cancer cells [164].

Organoids are confined to mimic organ-specific or tissue-specific micro-physiology, a constraint that should be considered before joining this promising new field. The lack of inter-organ communication is a major weakness of organoid systems. On the other hand, initiatives are already underway to circumvent this constraint. For instance, multiple organoids have been linked together to explore the gastrointestinal tract–liver–pancreas connection [165]. Despite the lingering difficulties, organoids retain significant promise for clinical translational research [166]. The scope of organoid technology has extended to include genetic manipulation, diverse omics, and drug-screening studies and a diversified co-culture system involving viruses, bacteria, and parasites. In the near future, the combination of organoid technology with single-cell transcriptomics will have a significant impact on personalized medicine.

## 5. Conclusions

The current study provides new and comprehensive data describing how LPS induces differential gene expression in intestinal organoids derived from dogs with chronic intestinal inflammation and small intestinal cancer. The crosstalk between the LPS/TLR4 signal transduction pathway and other metabolic pathways such as primary bile acid biosynthesis and secretion, peroxisome, renin–angiotensin system, glutathione metabolism, arachidonic acid, and Alzheimer’s disease pathways demonstrates that LPS plays a key role in chronic inflammation and carcinogenesis. In contrast, we observed contradictory effects of LPS including the increased proliferation and expression of several tumor-associated genes and the decreased expression of other cancer-associated genes in tumor enteroids including *LOC610994 (Zeta Crystallin)*, *Gpatch4*, *SLC7A1*, *ATP13A2*, and *TEX45*. In summary, this study may pave the way for the development of novel anti-inflammatory and anticancer therapeutics.

## Figures and Tables

**Figure 1 cancers-14-03525-f001:**
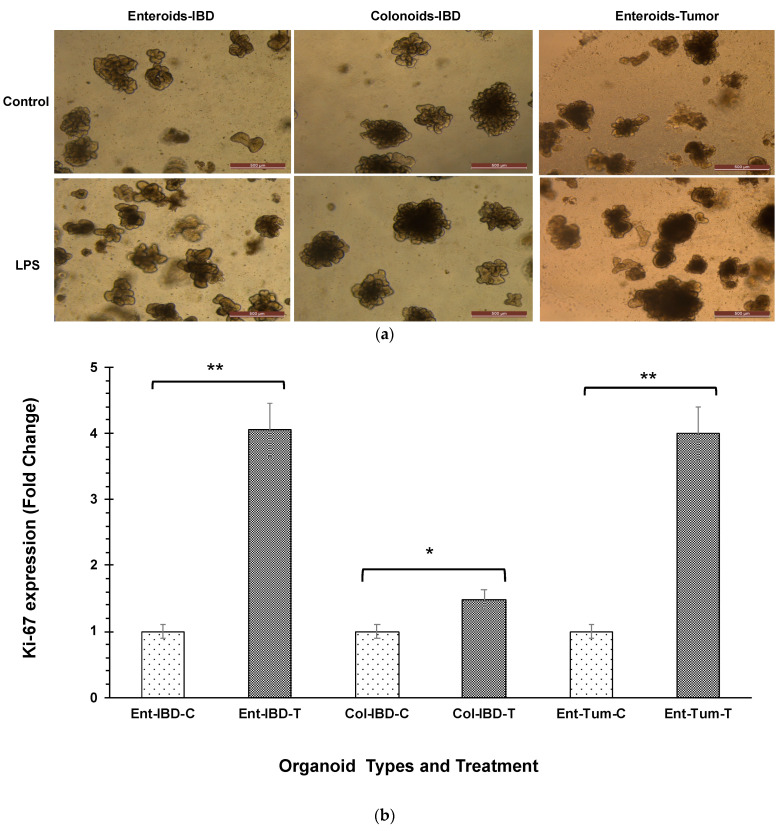
LPS stimulates higher proliferation. (**a**) Enteroids and colonoids from dogs with IBD and intestinal mast cell tumor, 48 h after LPS stimulation. Representative phase-contrast images of enteroids and colonoids after LPS stimulation. The control group received complete growth medium, whereas the LPS group received LPS 5 µg/mL in complete growth medium. (**b**) Expression of Ki-67 in enteroids and colonoids from dogs with IBD and intestinal mast cell tumor, 48 h after LPS stimulation as measured by qPCR. The control group received complete growth medium, whereas the LPS group received LPS 5 µg/mL in complete growth medium. GAPDH was used to normalize. Ent-IBD-C: control IBD enteroids; Ent-IBD-T: IBD enteroids following LPS stimulation; Col-IBD-C: control IBD colonoids; Col-IBD-T: IBD colonoids following LPS stimulation; Ent-Tum-C: control tumor enteroids; Ent-Tum-T: tumor enteroids following LPS stimulation. Histograms represent the mean Ki-67 expression (fold change) ± SD of four technical replicates used in the qPCR experiment. Unpaired two-tailed Student’s *t*-test was used for statistical analysis. * *p* < 0.0005, ** *p* < 0.00001. (**c**) Histograms represent induced expression levels (log-ratio M values) of proliferating cell nuclear antigen (*PCNA*), prominin 1 (*PROM1*), HOP homeobox (*HOPX*), and olfactomedin 4 (*OLFM4*) in LPS treated IBD colonoids vs. LPS treated IBD enteroids (Contrast-6), and LPS treated IBD enteroids vs. LPS treated tumor enteroids (Contrast-7). The log-ratio M values represent log(R/G) (log fold change) [44]. (**d**) Growth rate of organoids by LPS stimulation was evaluated by surface area and diameter of the organoids, crypt length, and the number of crypts per organoid. Data are presented as mean ± SD. Ent-IBD-C: control IBD enteroids; Ent-IBD-T: IBD enteroids following LPS stimulation; Col-IBD-C: control IBD colonoids; Col-IBD-T: IBD colonoids following LPS stimulation; Ent-Tum-C: control tumor enteroids; Ent-Tum-T: tumor enteroids following LPS stimulation. Unpaired two-tailed Student’s *t*-test was used for statistical analysis. * *p* < 0.05, ** *p* < 0.005, *** *p* < 0.0005.

**Figure 2 cancers-14-03525-f002:**
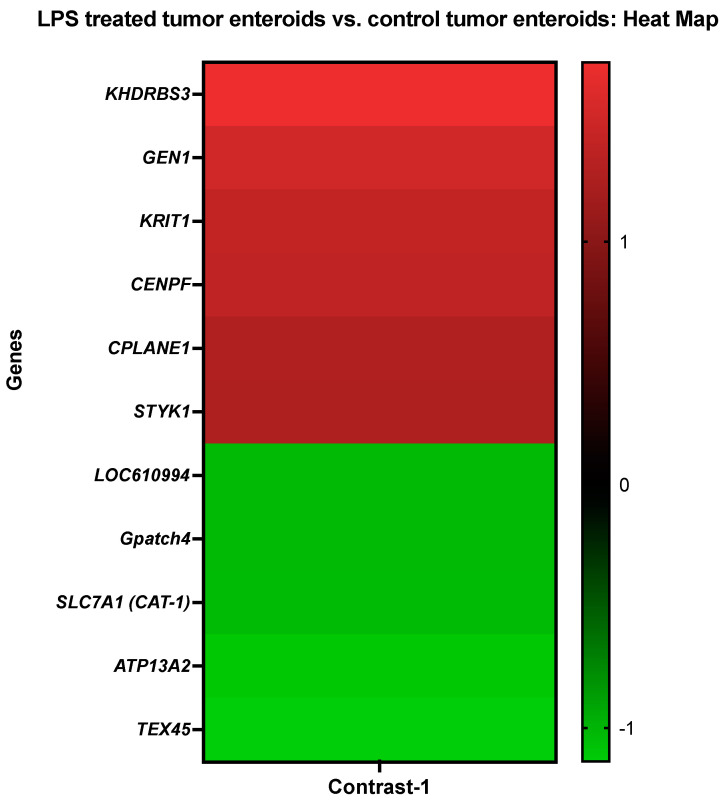
A heat map representing the color-coded expression levels (log-ratio M values) of the top six upregulated and five downregulated DEGs in the LPS treated tumor enteroids vs. the control tumor enteroids (Contrast-1). The log-ratio M values represent the log(R/G) (log fold change) [44]. The detailed information on genes is presented in Table 4.

**Figure 3 cancers-14-03525-f003:**
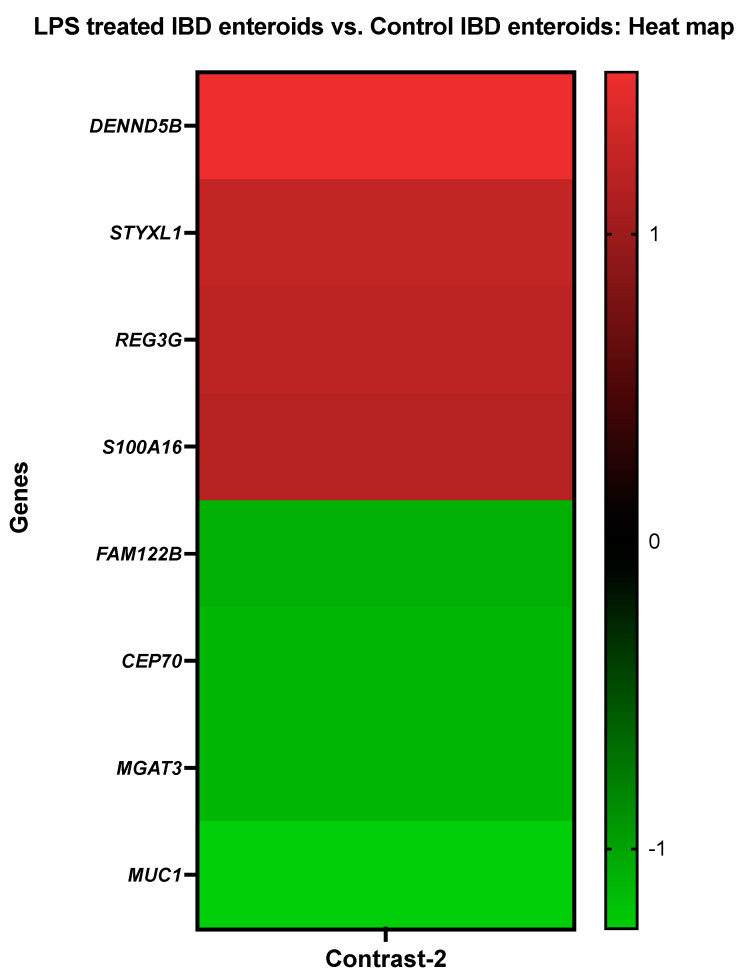
A heat map depicting the color-coded expression levels (log-ratio M values) of the top four upregulated and four downregulated DEGs in the LPS treated IBD enteroids vs. the control IBD enteroids (Contrast-2). The log-ratio M values represent the log(R/G) (log fold change) [44]. The detailed information on genes is presented in Table 5.

**Figure 4 cancers-14-03525-f004:**
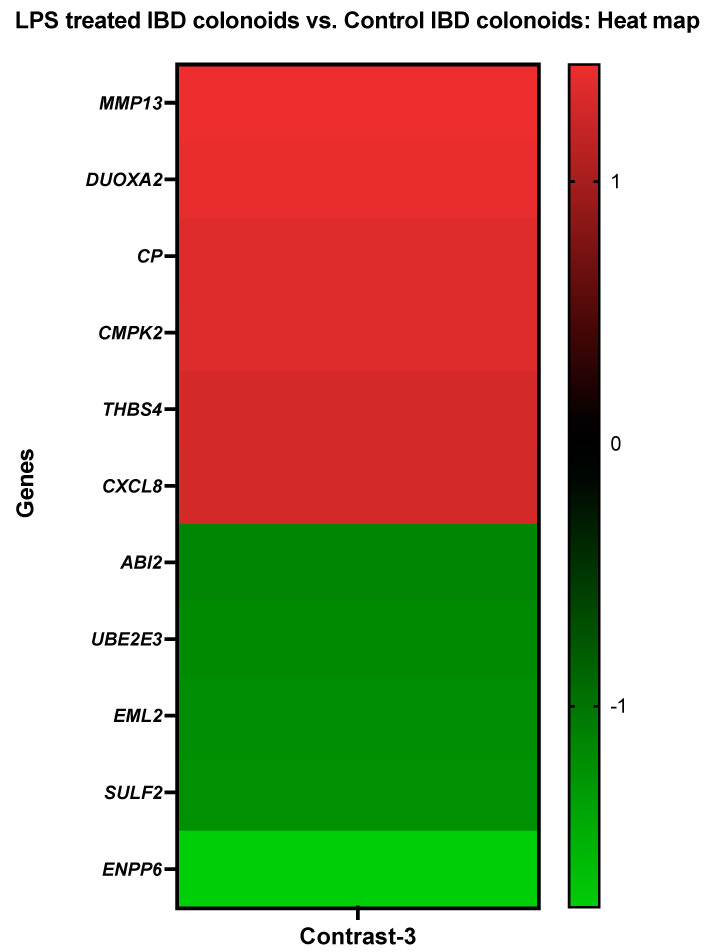
A heat map depicting the color-coded expression levels (log-ratio M values) of the top six upregulated and five downregulated DEGs in the LPS treated IBD colonoids vs. the control IBD colonoids (Contrast-3). The log-ratio M values represent the log(R/G) (log fold change) [44]. The detailed information on genes is presented in Table 6.

**Figure 5 cancers-14-03525-f005:**
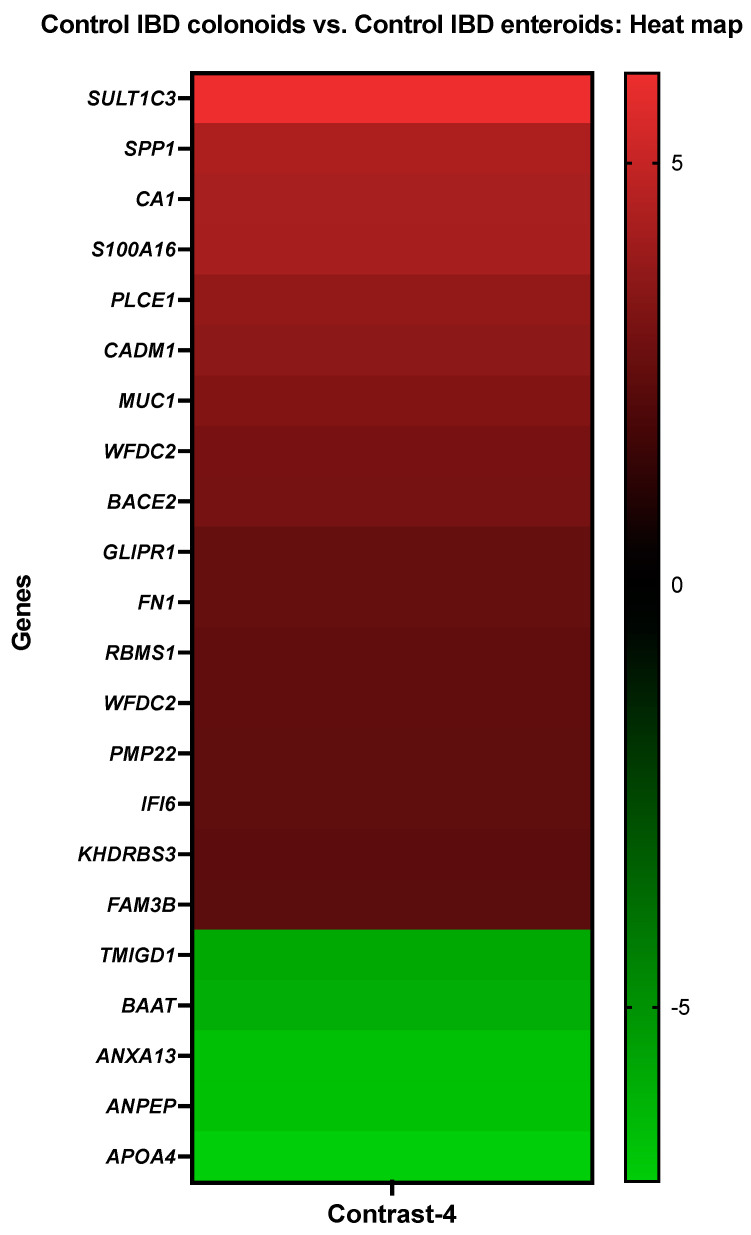
A heat map depicting the color-coded expression levels (log-ratio M values) of the top 17 upregulated and six downregulated DEGs in the control IBD colonoids vs. the control IBD enteroids (Contrast 4). The log-ratio M values represent the log(R/G) (log fold change) [44]. The detailed information on genes is presented in Table 7.

**Figure 6 cancers-14-03525-f006:**
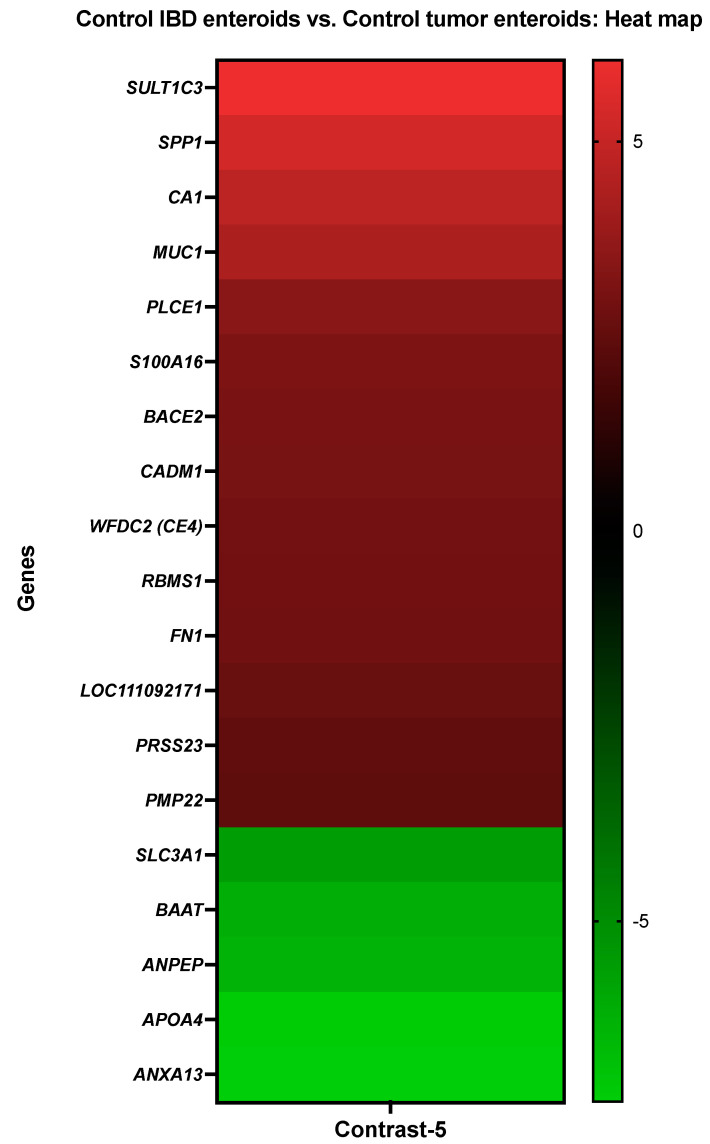
A heat map depicting the color-coded expression levels (log-ratio M values) of the top 14 upregulated and five downregulated DEGs in the control IBD enteroids vs. the control tumor enteroids (Contrast-5). The log-ratio M values represent the log(R/G) (log fold change) [44]. The detailed information on genes is presented in Table 8.

**Figure 7 cancers-14-03525-f007:**
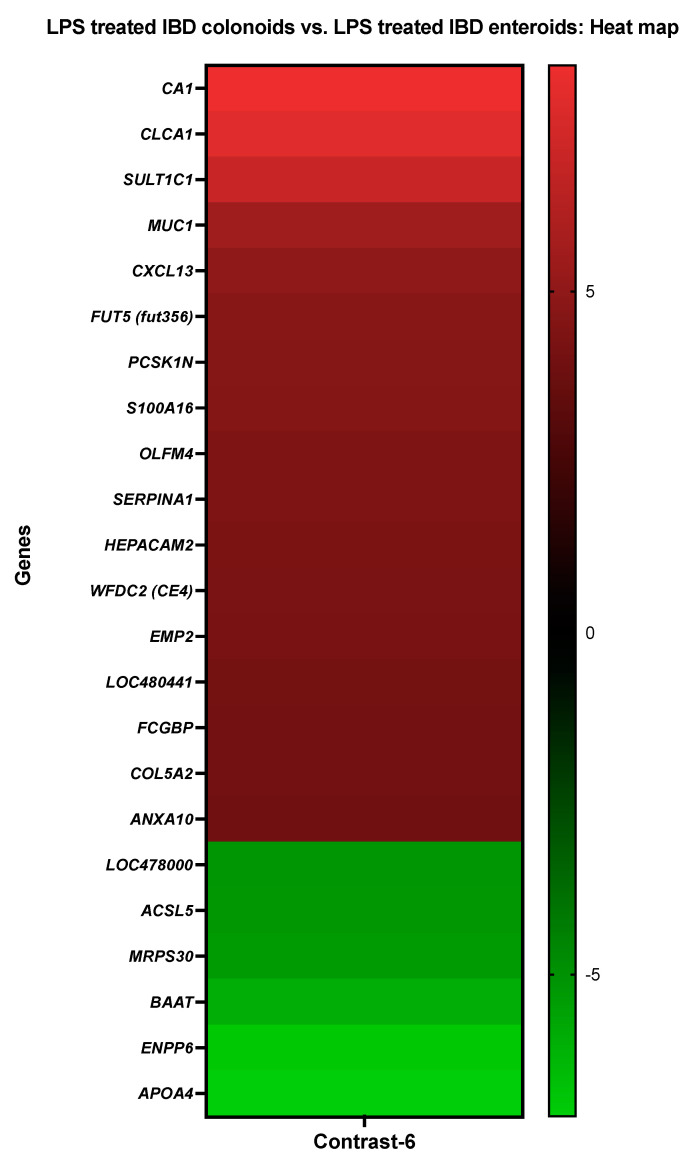
A heat map depicting the color-coded expression levels (log-ratio M values) of the top 17 upregulated and six downregulated DEGs in the LPS treated IBD colonoids vs. the LPS treated IBD enteroids (Contrast-6). The log-ratio M values represent the log(R/G) (log fold change) [44]. The detailed information on genes is presented in Table 9.

**Figure 8 cancers-14-03525-f008:**
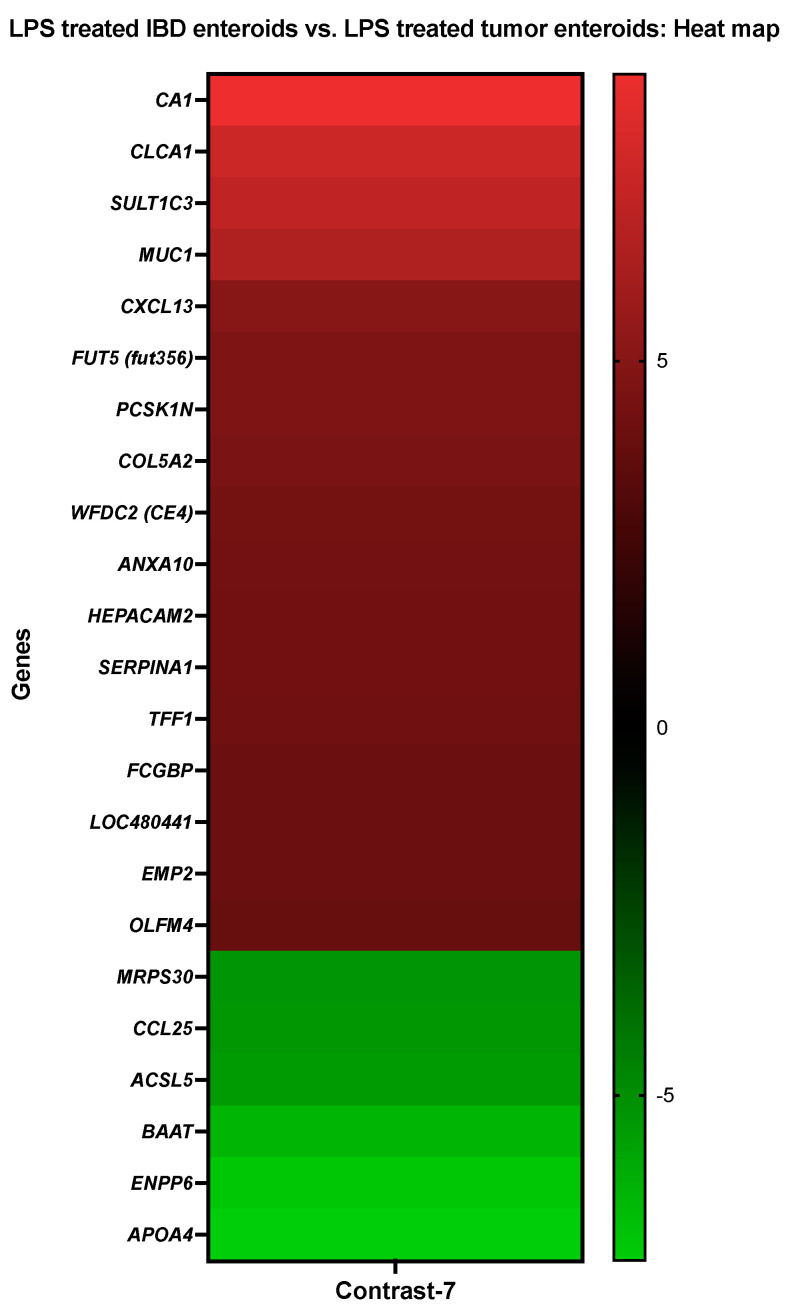
A heat map depicting the color-coded expression levels (log-ratio M values) of the top 17 upregulated and six downregulated DEGs in the LPS treated IBD enteroids vs. the LPS treated tumor enteroids (Contrast-7). The log-ratio M values represent the log(R/G) (log fold change) [44]. The detailed information on genes is presented in Table 10.

**Figure 9 cancers-14-03525-f009:**
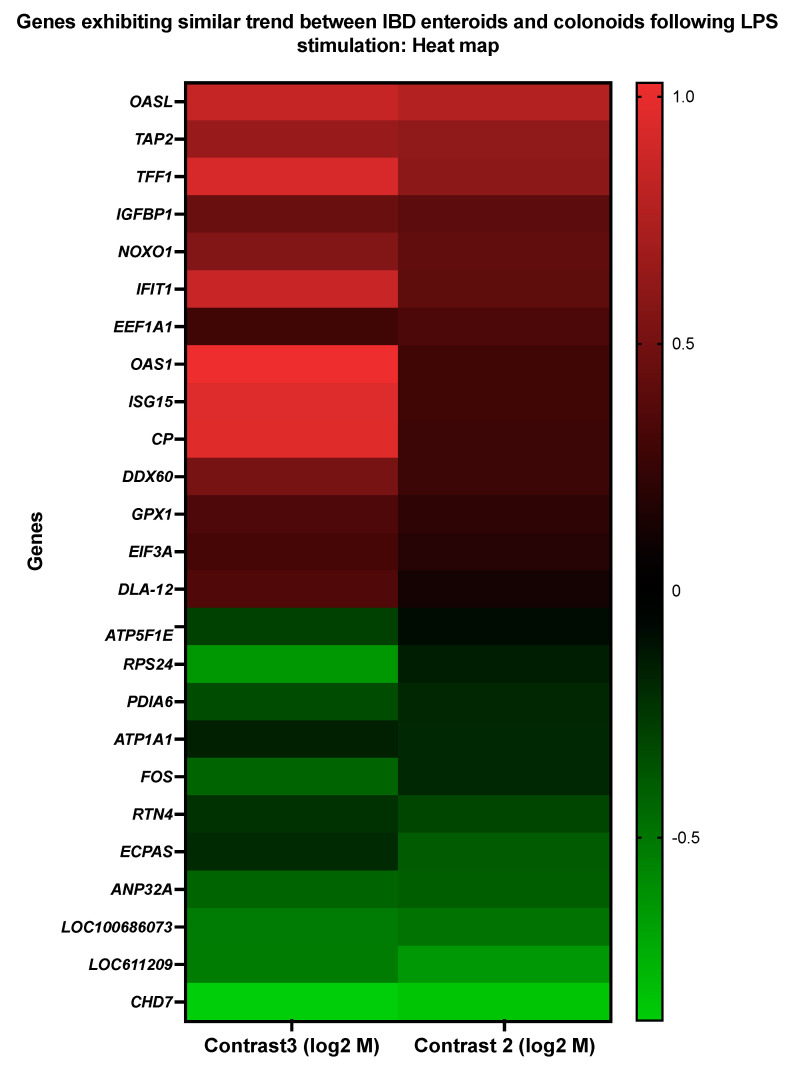
A heat map representing the color-coded expression levels (log-ratio *M* values) of 25 DEGs exhibiting a similar pattern of expression between the IBD enteroids and colonoids following LPS stimulation. The log-ratio *M* values represent the log(R/G) (log fold change) [44]. The detailed information on genes is presented in Table 11.

**Figure 10 cancers-14-03525-f010:**
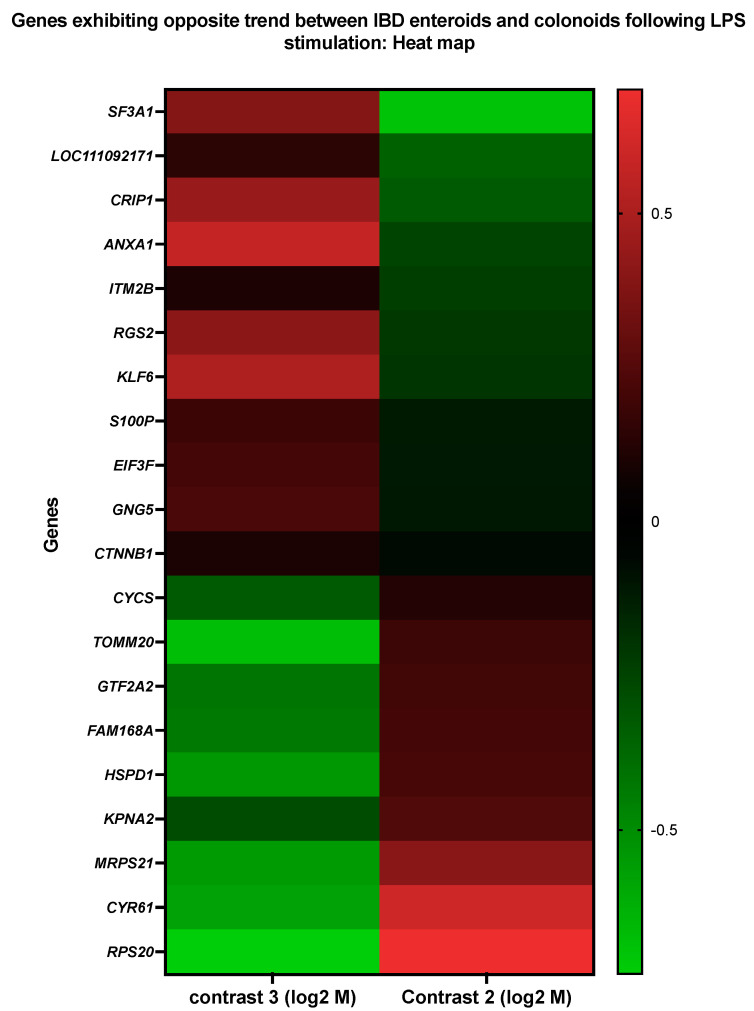
A heat map representing the color-coded expression levels (log-ratio M values) of DEGs upregulated in the IBD enteroids and downregulated in the IBD colonoids and vice versa between the IBD enteroids and colonoids following LPS stimulation. The log-ratio *M* values represent the log(R/G) (log fold change) [44]. The detailed information on genes is presented in Table 12.

**Figure 11 cancers-14-03525-f011:**
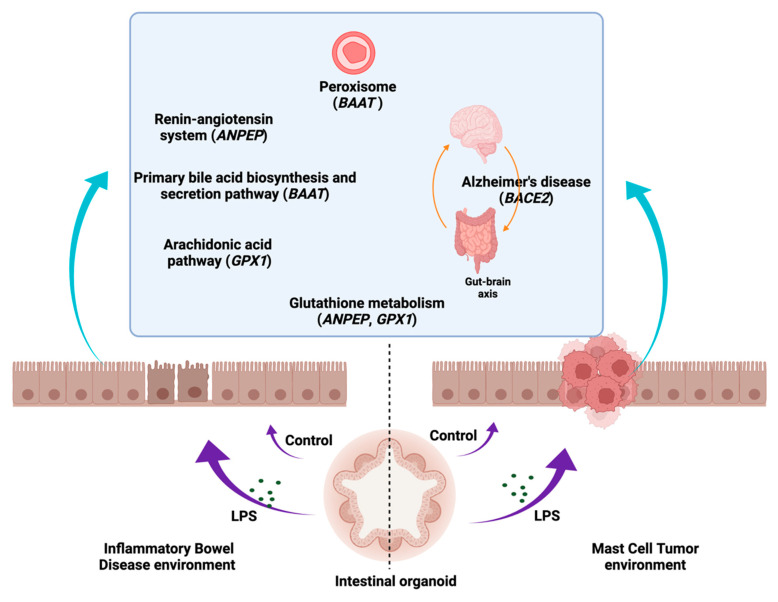
The association of several metabolic pathway genes in the IBD intestinal organoids and tumor enteroids. The figure was created with BioRender.com.

**Table 1 cancers-14-03525-t001:** The composition of complete chelating solution (CCS) and organoid culture media.

**Complete Chelating Solution (CCS)**		
**Composition**	**Concentration**	**Supplier**
Na_2_HPO_4_-2H_2_O	0.996 mg/mL	Sigma
KH_2_PO_4_	1.08 mg/mL	Sigma
NaCl	5.6 mg/mL	Fisher Chemical
KCl	0.12 mg/mL	Fisher Chemical
Sucrose	15 mg/mL	Fisher Chemical
D-Sorbitol	10 mg/mL	Fisher Chemical
Dithiothreitol (DTT)	520 μM	Promega
Pen Strep	Penicillin: 196 units/mL; Streptomycin: 196 μg/mL	Gibco
**Incomplete Media without an ISC Growth Factor (IMGF-)**		
**Composition**	**Concentration**	**Supplier**
Advanced DMEM/F12	Base Media	Gibco
FBS	8%	Corning
GlutaMAX™ Supplement (200 mM L-alanyl-L-glutamine dipeptide in 0.85% NaCl)	2 mM	Gibco
HEPES	10 mM	VWR Life Science
Primocin	100 µg/mL	InvivoGen
**Complete Media with ISC Growth Factors (CMGF+)**		
**Composition**	**Concentration**	**Supplier**
Advanced DMEM/F12	Base Media	Gibco
FBS	8%	Corning
GlutaMAX™ Supplement (200 mM L-alanyl-L-glutamine dipeptide in 0.85% NaCl)	2 mM	Gibco
HEPES	10 mM	VWR Life Science
Primocin	100 µg/mL	InvivoGen
B27 Supplement	1x	Gibco
N2 Supplement	1x	Gibco
N-Acetyl-L-cysteine	1 mM	Sigma
Murine EGF	50 ng/mL	PeproTech
Murine Noggin	100 ng/mL	PeproTech
Human R-Spondin-1	500 ng/mL	PeproTech
Murine Wnt-3a	100 ng/mL	PeproTech
[Leu^15^]-Gastrin I human	10 nM	Sigma
Nicotinamide	10 mM	Sigma
TGF beta 1 Recombinant Protein	500 nM	ProSci
SB202190 (P38 inhibitor)	10 µM	Sigma
TMS (trimethoprim sulfate)	10 µg/mL	Sigma
* ROCK inhibitor (Y-27632)	10 µM	EMD Millipore Corp.
* Aminopyrimidine CHIR99021 (GSK3β inhibitor)	2.5 µM	Reprocell

* The CMGF + medium with ROCK-I inhibitor and glycogen synthase kinase 3β inhibitor was used for the first two days of ISC culture to enhance ISC survival and prevent apoptosis.

**Table 2 cancers-14-03525-t002:** A detailed description of the samples.

Description of the Sample	Sample Designation
Control enteroids from the MC tumor environment (9 biological replicates (wells) pooled together)	Sample-1
LPS treated enteroids from the MC tumor environment (9 biological replicates (wells) pooled together)	Sample-2
Control enteroids from IBD dogs (9 biological replicates (wells) pooled together)	Sample-3
LPS treated enteroids from IBD dogs (9 biological replicates (wells) pooled together)	Sample-4
Control colonoids from IBD dogs (9 biological replicates (wells) pooled together)	Sample-5
LPS treated colonoids from IBD dogs (9 biological replicates (wells) pooled together)	Sample-6

MC = mast cell.

**Table 3 cancers-14-03525-t003:** The differentially expressed genes within the enteroid and colonoid groups.

Serial No	Group Comparison	Total Number of Differentially Expressed Genes	Upregulated Genes (%)	Downregulated Genes (%)
1 (2–1)	LPS treated tumor enteroids vs. Control tumor enteroids	677	56	44
2 (4–3)	LPS treated IBD enteroids vs. Control IBD enteroids	639	49	51
3 (6–5)	LPS treated IBD colonoids vs. Control IBD colonoids	684	48	52
4 (5–3)	Control IBD colonoids vs. Control IBD enteroids	411	44	56
5 (3–1)	Control IBD enteroids vs. Control tumor enteroids	376	49	51
6 (6–4)	LPS treated IBD colonoids vs. LPS treated IBD enteroids	421	39	61
7 (4–2)	LPS treated IBD enteroids vs. LPS treated tumor enteroids	314	51	49

**Table 4 cancers-14-03525-t004:** Gene Ontology (GO) analysis and details about the top six upregulated and five downregulated DEGs in the LPS treated tumor enteroids vs. the control tumor enteroids.

**Details for Top Upregulated and Downregulated DEGs**		
**Gene Description**	**Gene Symbol**	**Probe Set ID**
KH domain-containing, RNA-binding, signal transduction-associated protein 3	*KHDRBS3*	CfaAffx.2644.1.S1_at
Flap endonuclease GEN homolog 1	*GEN1*	CfaAffx.6562.1.S1_at
Krev interaction trapped protein 1	*KRIT1*	CfaAffx.3865.1.S1_s_at
Centromere protein F	*CENPF*	CfaAffx.19540.1.S1_at
Ciliogenesis and planar polarity effector complex subunit 1	*CPLANE1*	Cfa.1768.1.A1_at
Tyrosine-protein kinase STYK1	*STYK1*	CfaAffx.20747.1.S1_s_at
Quinone oxidoreductase-like protein 2/Crystallin Zeta Like 2	*LOC610994*	Cfa.3282.1.S1_s_at
G patch domain-containing protein 4 isoform X1	*Gpatch4*	CfaAffx.25660.1.S1_at
High affinity cationic amino acid transporter 1	*SLC7A1* (*CAT-1*)	CfaAffx.10937.1.S1_at
Cation-transporting ATPase 13A2	*ATP13A2*	CfaAffx.24286.1.S1_at
Testis expressed 45	*TEX45*	CfaAffx.27913.1.S1_at
**Gene Ontology (GO) Analysis of 11 DEGs (Annotated for Biological Process)**		
**GO-Terms**	**Number of Genes**	**Genes**
Cellular metabolic process and regulation, organic substance metabolic process, nitrogen compound metabolic process (18%)	5	*CENPF*, *GEN1*, *STYK1*, *KHDRBS3*, *ATP13A2*
Regulation of metabolic process (11%)	3	*CENPF*, *KHDRBS3*, *ATP13A2*
Establishment of localization (11%)	3	*CENPF*, *ATP13A2*, *SLC7A1/CAT-1*
Macromolecule localization, cellular localization (7%)	2	*CENPF*, *ATP13A2*
Regulation of molecular function, cellular homeostasis, regulation of biological quality (7%)	2	*KRIT1*, *ATP13A2*
Biosynthetic process, cell cycle process, negative regulation of cellular process, chromosome segregation (7%)	2	*CENPF*, *GEN1*
Positive regulation of cellular process, cellular response to stimulus, cellular component organization or biogenesis, response to stress (7%)	2	*GEN1*, *ATP13A2*
Anatomical structure development, regulation of developmental process, multicellular organism development (7%)	2	*CENPF*, *KRIT1*
Transmembrane transport (7%)	2	*ATP13A2*, *SLC7A1/CAT-1*
Positive regulation of transport, export from cell, response to chemical, regulation of localization, catabolic process, process utilizing autophagic mechanism, positive regulation of metabolic process, positive regulation of establishment of protein localization, vesicle-mediated transport (4%)	1	*ATP13A2*
Negative regulation of metabolic process (4%)	1	*CENPF*
Cell cycle checkpoint, microtubule-based process (4%)	1	*GEN1*
Anatomical structure formation involved in morphogenesis, Negative regulation of developmental process, regulation of multicellular organismal process, establishment or maintenance of cell polarity, negative regulation of multicellular organismal process, anatomical structure morphogenesis (4%)	1	*KRIT1*
Oxidation-reduction process (4%)	1	*LOC610994*
**Gene Ontology (GO) Analysis of 11 DEGs (Annotated for Molecular Function)**		
**GO-Terms**	**Number of Genes**	**Genes**
Identical protein binding (14%)	3	*CENPF*, *GEN1*, *KHDRBS3*
Nucleic acid binding (14%)	3	*Gpatch4*, *GEN1*, *KHDRBS3*
Nucleoside phosphate binding, anion binding, ribonucleotide binding (10%)	2	*STYK1*, *ATP13A2*
Protein dimerization activity (10%)	2	*CENPF*, *GEN1*
Cation binding (10%)	2	*GEN1*, *ATP13A2*
Nucleotide binding (10%)	2	*STYK1*, *ATP13A2*
Inorganic molecular entity transmembrane transporter activity (5%)	1	*SLC7A1 (CAT-1)*
Protein kinase activity, transferase activity, transferring phosphorus-containing groups (5%)	1	*STYK1*
Transcription factor binding, protein C-terminus binding, dynein complex binding, cytoskeletal protein binding (5%)	1	*CENPF*
Phospholipid binding, hydrolase activity, acting on acid anhydrides (5%)	1	*ATP13A2*
Hydrolase activity, acting on ester bonds, deoxyribonuclease activity (5%)	1	*GEN1*
Protein domain specific binding (5%)	1	*KHDRBS3*
Ion transmembrane transporter activity (5%)	1	*SLC7A1 (CAT-1)*
**Gene Ontology (GO) Analysis of 11 DEGs (Annotated for Cellular Component)**		
**GO-Terms**	**Number of Genes**	**Genes**
Intracellular organelle (17%)	5	*CENPF*, *GEN1*, *KRIT1*, *KHDRBS3*, *ATP13A2*
Intracellular non-membrane-bounded organelle (10%)	3	*CENPF*, *GEN1*, *KRIT1*
Membrane-bounded organelle (10%)	3	*CENPF*, *KHDRBS3*, *ATP13A2*
Integral component of membrane (10%)	3	*STYK1*, *ATP13A2*, *SLC7A1 (CAT-1)*
Non-membrane-bounded organelle (10%)	3	*CENPF*, *GEN1*, *KRIT1*
Intracellular organelle lumen (7%)	2	*CENPF*, *KHDRBS3*
Centrosome (7%)	2	*CENPF*, *GEN1*
Plasma membrane bounded cell projection (7%)	2	*CENPF*, *ATP13A2*
Organelle lumen (7%)	2	*CENPF*, *KHDRBS3*
Plasma membrane (7%)	2	*STYK1*, *SLC7A1 (CAT-1)*
Intrinsic component of organelle membrane, endosome, organelle membrane, intracellular vesicle, vacuole, neuronal cell body, cytoplasmic vesicle, transport vesicle (3%)	1	*ATP13A2*
Cilium, ciliary basal body, cytoplasmic region, organelle envelope, kinetochore, chromosome, centromeric region, nuclear envelope (3%)	1	*CENPF*

**Table 5 cancers-14-03525-t005:** The Gene Ontology (GO) analysis and details about the top four upregulated and four downregulated DEGs in the LPS treated IBD enteroids vs. the control IBD enteroids. The DEGs annotated for the biological process, molecular function, and cellular component categories are shown. The percentages represent the number of genes in each functional category.

**Details for Top Upregulated and Downregulated DEGs**		
**Gene Description**	**Gene Symbol**	**Probe Set ID**
DENN domain containing 5B	*DENND5B*	Cfa.19506.1.S1_at
Serine/threonine/tyrosine-interacting-like protein 1	*STYXL1*	Cfa.17400.1.S1_s_at
Regenerating islet-derived protein 3-gamma-like	*REG3G*	Cfa.1742.1.S1_at
Protein S100-A16	*S100A16*	Cfa.18759.2.S1_at
Protein FAM122B	*FAM122B*	CfaAffx.28830.1.S1_s_at
Centrosomal protein of 70 kda	*CEP70*	Cfa.20146.1.S1_s_at
Beta-1,4-mannosyl-glycoprotein 4-beta-N-acetylglucosaminyltransferase	*MGAT3*	Cfa.9863.1.A1_at
Mucin 1	*MUC1*	CfaAffx.26061.1.S1_at
**Gene Ontology (GO) Analysis of DEGs (Annotated for Biological Process)**		
**GO-Terms**	**Genes**	
Organonitrogen compound metabolic process (25%)	*STYXL1*	
Microtubule cytoskeleton organization, Regulation of microtubule-based process (25%)	*CEP70*	
Cellular component biogenesis/organization and regulation (25%)	*CEP70*	
Phosphorus/cellular macromolecule/protein metabolic process (25%)	*STYXL1*	
**Gene Ontology (GO) Analysis of DEGs (Annotated for Molecular Function)**		
**GO-Terms**	**Genes**	
Phosphoprotein phosphatase activity (25%)	*STYXL1*	
Identical protein binding, cytoskeletal protein binding (25%)	*CEP70*	
Hydrolase activity, acting on ester bonds (25%)	*STYXL1*	
Cation binding (25%)	*S100A16*	
**Gene Ontology (GO) Analysis of DEGs (Annotated for Cellular Component)**		
**GO-Terms**	**Genes**	
Intracellular organelle (29%)	*MUC1*, *CEP70*	
Non-membrane-bounded organelle (14%)	*CEP70*	
Intracellular non-membrane-bounded organelle (14%)	*CEP70*	
Centrosome (14%)	*CEP70*	
Membrane/intrinsic component of membrane, cytoplasm (14%)	*MUC1*	
Cell periphery/apical part of cell (14%)	*MUC1*	

**Table 6 cancers-14-03525-t006:** The Gene Ontology (GO) analysis and details of the DEGs in the LPS treated IBD colonoids vs. the control IBD colonoids annotated for the biological process, molecular function, and cellular component categories. The percentages represent the number of genes in each functional category.

**Details for Upregulated and Downregulated DEGs**		
**Gene Description**	**Gene Symbol**	**Probe Set ID**
Collagenase 3	*MMP13*	CfaAffx.23153.1.S1_at
Dual oxidase maturation factor 2	*DUOXA2*	CfaAffx.21162.1.S1_at
Ceruloplasmin	*CP*	CfaAffx.13209.1.S1_s_at
UMP-CMP kinase 2, mitochondrial	*CMPK2*	Cfa.19154.1.S1_a_at
Thrombospondin-4	*THBS4*	CfaAffx.14209.1.S1_s_at
Interleukin-8	*CXCL8*	Cfa.3510.1.S1_s_at
abl Interactor 2	*ABI2*	Cfa.1679.1.A1_at
Ubiquitin-conjugating enzyme E2 E3	*UBE2E3*	CfaAffx.805.1.S1_s_at
Echinoderm microtubule-associated protein-like 2	*EML2*	Cfa.9105.1.A1_s_at
Extracellular sulfatase Sulf-2-like	*SULF2*	Cfa.6393.1.A1_at
Ectonucleotide pyrophosphatase/phosphodiesterase family member 6	*ENPP6*	Cfa.6996.1.A1_at
**Gene Ontology (GO) Analysis of DEGs (Annotated for Biological Process)**		
**GO-Terms**	**Genes**	
Primary metabolic process, organic substance/nitrogen compound metabolic process (14%)	*CMPK2*, *ENPP6*, *MMP13*	
Cellular metabolic process (9%)	*CMPK2*, *MMP13*	
Establishment of localization (9%)	*DUOXA2*, *CP*	
Catabolic process (9%)	*ENPP6*, *MMP13*	
Biosynthetic process, small molecule metabolic process (5%)	*CMPK2*	
Regulation of biological quality, cellular homeostasis, oxidation-reduction process (5%)	*CP*	
Taxis (5%)	*CXCL8*	
Movement of cell or subcellular component, cell motility (5%)	*CXCL8*	
Localization of cell, cell activation (5%)	*CXCL8*	
Cell communication, signal transduction, response to chemical/external stimulus, cellular response to stimulus, immune response (5%)	*CXCL8*	
Leukocyte activation/migration (5%)	*CXCL8*	
Regulation of cellular process, response to stress (5%)	*CXCL8*	
Macromolecule localization (5%)	*DUOXA2*	
Developmental growth, ossification, collagen metabolic process (5%)	*MMP13*	
Cellular component organization or biogenesis (5%)	*MMP13*	
Multicellular organism development, anatomical structure development/morphogenesis/organ growth, biomineral tissue development (5%)	*MMP13*	
Cell adhesion (5%)	*THBS4*	
**Gene Ontology (GO) Analysis of DEGs (Annotated for Molecular Function)**		
**GO-Terms**	**Genes**	
Metal ion binding (19%)	*THBS4*, *CP*, *MMP13*	
Phosphotransferase activity, phosphate group as acceptor (6%)	*CMPK2*	
Purine ribonucleoside triphosphate binding (6%)	*UBE2E3*	
Metallopeptidase activity (6%)	*MMP13*	
G protein-coupled receptor binding (6%)	*CXCL8*	
Endopeptidase activity (6%)	*MMP13*	
Receptor ligand activity (6%)	*CXCL8*	
Cytokine receptor binding (6%)	*CXCL8*	
Purine nucleotide binding (6%)	*UBE2E3*	
Phosphoric ester hydrolase activity (6%)	*ENPP6*	
Purine ribonucleotide binding (6%)	*UBE2E3*	
Sulfuric ester hydrolase activity (6%)	*SULF2*	
Kinase activity (6%)	*CMPK2*	
Oxidoreductase activity, oxidizing metal ions, oxygen as acceptor (6%)	*CP*	
**Gene Ontology (GO) Analysis of DEGs (Annotated for Cellular Component)**		
**GO-Terms**	**Genes**	
Extracellular region (28%)	*CXCL8*, *THBS4*, *ENPP6*, *CP*, *MMP13*	
Membrane (17%)	*EML2*, *DUOXA2*, *ENPP6*	
Extracellular space (11%)	*CXCL8*, *CP*	
Intrinsic component of membrane (11%)	*EML2*, *DUOXA2*	
Cytoplasm (6%)	*DUOXA2*	
Intracellular anatomical structure (6%)	*DUOXA2*	
Organelle, organelle subcompartment (6%)	*DUOXA2*	
Cell periphery (6%)	*ENPP6*	
Extracellular matrix (6%)	*MMP13*	
Endomembrane system (6%)	*DUOXA2*	

**Table 7 cancers-14-03525-t007:** The Gene Ontology (GO) analysis and details about the top 17 upregulated and six downregulated DEGs in the control IBD colonoids vs. the control IBD enteroids. The DEGs annotated for biological process, molecular function, and cellular component categories are presented. The number of DEGs that fell into each of these categories is shown in percentages.

**Details about DEGs in Control IBD Colonoids vs. Control IBD Enteroids**		
**Gene Description**	**Gene Symbol**	**Probe Set ID**
Sulfotransferase family 1C member 3	*SULT1C3*	CfaAffx.4011.1.S1_at
Osteopontin	*SPP1*	Cfa.9240.1.S1_at
Carbonic anhydrase 1	*CA1*	Cfa.6413.1.A1_at/CfaAffx.13785.1.S1_at
Protein S100-A16	*S100A16*	Cfa.18759.2.S1_at
Phospholipase C epsilon 1	*PLCE1*	Cfa.12506.3.S1_at
Cell adhesion molecule 1	*CADM1*	Cfa.11274.1.A1_at/Cfa.10739.1.A1_s_at
Mucin-1	*MUC1*	CfaAffx.26061.1.S1_at/Cfa.7074.1.A1_at
WAP four-disulfide core domain protein 2	*WFDC2*	CfaAffx.15227.1.S1_s_at
beta-Secretase 2	*BACE2*	Cfa.20396.1.S1_at
Glioma pathogenesis-related protein 1	*GLIPR1*	Cfa.5134.1.A1_s_at
ENSCAFT00000022799 XM_014110988.1 PREDICTED: Canis lupus familiaris fibronectin 1	*FN1*	CfaAffx.22155.1.S1_s_at
RNA-binding motif, single-stranded-interacting protein 1 isoform X3	*RBMS1*	Cfa.5781.1.A1_s_at
WAP four-disulfide core domain protein 2	*WFDC2*	Cfa.3780.1.S1_at
Peripheral myelin protein 22	*PMP22*	CfaAffx.27421.1.S1_at
Interferon alpha-inducible protein 6	*IFI6*	Cfa.20456.1.S1_s_at
KH domain-containing, RNA-binding, signal transduction-associated protein 3	*KHDRBS3*	Cfa.334.1.A1_s_at
Protein FAM3B	*FAM3B*	Cfa.12168.1.A1_s_at
Transmembrane and immunoglobulin domain-containing protein 1	*TMIGD1*	CfaAffx.29036.1.S1_at
Bile acid-CoA:amino acid N-acyltransferase-like	*BAAT*	CfaAffx.4748.1.S1_at
Annexin A13	*ANXA13*	Cfa.3796.1.A1_s_at
Aminopeptidase N	*ANPEP*	Cfa.3774.1.A1_s_at, Cfa.20798.1.S1_at
Apolipoprotein A-IV	*APOA4*	Cfa.6294.1.A1_at
**Gene Ontology (GO) Analysis of DEGs (Annotated for Biological Process)**		
**GO-Terms**	**Genes**	
Cellular metabolic process (7%)	*CA1*, *ANPEP*, *PLCE1*, *BAAT*, *APOA4*, *WFDC2*	
Regulation of cellular process (6%)	*ANXA13*, *ANPEP*, *TMIGD1*, *APOA4*, *WFDC2*	
Primary metabolic process, organic substance/nitrogen compound metabolic process (6%)	*ANPEP*, *BAAT*, *APOA4*, *BACE2*, *WFDC2*	
Regulation of biological quality (5%)	*FAM3B*, *ANPEP*, *TMIGD1*, *APOA4*	
Negative regulation of cellular process (4%)	*ANXA13*, *TMIGD1*, *WFDC2*	
Small molecule metabolic process (4%)	*CA1*, *BAAT*, *APOA4*	
Establishment of localization, macromolecule localization (4%)	*ANXA13*, *FAM3B*, *APOA4*	
Cell communication (4%)	*FAM3B*, *ANPEP*, *PLCE1*	
Regulation of localization (4%)	*ANXA13*, *TMIGD1*, *APOA4*	
Cell death, apoptotic process (4%)	*TMIGD1*, *PMP22*, *CADM1*	
Anatomical structure development/morphogenesis (4%)	*ANPEP*, *PMP22*	
Response to stress, regulation of metabolic process/molecular function/hydrolase activity (2%)	*APOA4*, *WFDC2*	
System process, catabolic process (2%)	*ANPEP*, *APOA4*	
Signal transduction, cellular response to stimulus (2%)	*ANPEP*, *PLCE1*	
Multicellular organism development, cellular developmental process, anatomical structure formation involved in morphogenesis (2%)	*ANPEP*, *PMP22*	
Positive regulation of cellular process (2%)	*ANXA13*, *APOA4*	
Cellular component organization or biogenesis (2%)	*APOA4*, *PMP22*	
Biosynthetic process (2%)	*BAAT*, *APOA4*	
Cell adhesion (2%)	*SPP1*, *CADM1*	
Biological process involved in symbiotic interaction (1%)	*ANPEP*	
Positive/negative regulation of transport, vesicle-mediated transport (1%)	*ANXA13*	
Cellular localization, positive/negative regulation of establishment of protein localization (1%)	*ANXA13*	
Plasma lipoprotein particle organization, regulation of plasma lipoprotein particle levels (1%)	*APOA4*	
Response to chemical, digestion, regulation of multicellular organismal process (1%)	*APOA4*	
Positive regulation of metabolic process/catalytic activity (1%)	*APOA4*	
Cell–cell signaling, export from cell (1%)	*FAM3B*	
Ensheathment of neurons (1%)	*PMP22*	
Ossification (1%)	*SPP1*	
Cell population proliferation (1%)	*TMIGD1*	
Regulation of locomotion, cell motility (1%)	*TMIGD1*	
Localization of cell, movement of cell or subcellular component (1%)	*TMIGD1*	
Response to external stimulus/biotic stimulus/other organism (1%)	*WFDC2*	
Immune response (1%)	*WFDC2*	
Negative regulation of metabolic process/catalytic activity (1%)	*WFDC2*	
Calcium-mediated signaling, positive regulation of cytosolic calcium ion concentration, epidermal growth factor receptor signaling pathway, Ras protein signal transduction, regulation of Ras protein signal transduction/G protein-coupled receptor signaling pathway, inositol phosphate-mediated signaling, phosphatidylinositol-mediated signaling, phospholipase C-activating G protein-coupled receptor signaling pathway (1%)	*PLCE1*	
Lipid catabolic process, diacylglycerol biosynthetic process (1%)	*PLCE1*	
Regulation of protein kinase activity, positive regulation of lamellipodium assembly/MAPK cascade (1%)	*PLCE1*	
Regulation of cell growth, cytoskeleton organization (1%)	*PLCE1*	
Heart development, glomerulus development, regulation of smooth muscle contraction (1%)	*PLCE1*	
Liver/brain development, spermatogenesis (1%)	*CADM1*	
Cell differentiation/recognition (1%)	*CADM1*	
Detection of stimulus (1%)	*CADM1*	
Homophilic cell adhesion/heterophilic cell–cell adhesion via plasma membrane cell adhesion molecules (1%)	*CADM1*	
Immune system process, positive regulation of cytokine production/natural killer cell mediated cytotoxicity (1%)	*CADM1*	
Susceptibility to natural killer cell mediated cytotoxicity (1%)	*CADM1*	
**Gene Ontology (GO) Analysis of DEGs (Annotated for Molecular Function)**		
**GO-Terms**	**Genes**	
Metal ion binding (21%)	*ANXA13*, *CA1*, *ANPEP*, *ANPEP*, *CA1*	
RNA binding (8%)	*RBMS1*, *KHDRBS3*	
Metallopeptidase activity (4%)	*ANPEP*	
Peptidase inhibitor activity, endopeptidase regulator activity (4%)	*WFDC2*	
Exopeptidase activity (4%)	*ANPEP*	
Transferase activity, transferring acyl groups other than amino-acyl groups (4%)	*BAAT*	
Aspartic-type peptidase activity, endopeptidase activity (4%)	*BACE2*	
Calcium-dependent phospholipid binding, phosphatidylglycerol binding (4%)	*ANXA13*	
Sterol binding (4%)	*APOA4*	
Thiolester hydrolase activity (4%)	*BAAT*	
SH3 Domain binding (4%)	*KHDRBS3*	
Sulfotransferase activity (4%)	*SULT1C3*	
Hydrolase/carboxylic ester hydrolase activity (4%)	*CA1*	
Enzyme binding, metal ion binding (4%)	*PLCE1*	
Phosphatidylinositol phospholipase C activity, small GTPase binding, guanyl-nucleotide exchange factor activity (4%)	*PLCE1*	
Cholesterol binding (4%)	*APOA4*	
Phosphatidylcholine-sterol O-acyltransferase activator activity (4%)	*APOA4*	
Signaling receptor binding (4%)	*CADM1*	
Cell adhesion molecule binding, PDZ domain binding, protein homodimerization activity (4%)	*CADM1*	
**Gene Ontology (GO) Analysis of DEGs (Annotated for Cellular Component)**		
**GO-Terms**	**Genes**	
Intracellular anatomical structure (14%)	*ANXA13*, *MUC1*, *CA1*, *ANPEP*, *PLCE1*, *BAAT*, *TMIGD1*, *BACE2*, *KHDRBS3*	
Membrane, cell membrane (14%)	*ANXA13*, *MUC1*, *ANPEP*, *PLCE1*, *TMIGD1*, *GLIPR1*, *BACE2*, *PMP22*, *CADM1*, *IFI6*	
Cytoplasm and cytosol (12%)	*ANXA13*, *CA1*, *MUC1*, *ANPEP*, *PLCE1*, *BAAT*, *TMIGD1*, *BACE2*	
Intrinsic component of membrane (12%)	*MUC1*, *ANPEP*, *TMIGD1*, *GLIPR1*, *CADM1*, *BACE2*, *PMP22*, *IFI6*	
Extracellular region (11%)	*FAM3B*, *ANPEP*, *PLCE1*, *SPP1*, *APOA4*, *GLIPR1*, *WFDC2*	
Organelle (11%)	*ANXA13*, *MUC1*, *ANPEP*, *PLCE1*, *BAAT*, *BACE2*, *KHDRBS3*	
Cell periphery (9%)	*ANXA13*, *MUC1*, *ANPEP*, *TMIGD1*, *BACE2*, *PMP22*	
Extracellular space (8%)	*ANPEP*, *PLCE1*, *GLIPR1*, *APOA4*, *WFDC2*	
Endomembrane system (5%)	*ANXA13*, *PLCE1*, *BACE2*	
Apical part of cell (3%)	*ANXA13*, *MUC1*	
Golgi apparatus membrane, lamellipodium (2%)	*PLCE1*	
Basolateral plasma membrane, cell–cell junction, neuron projection, postsynaptic density (2%)	*CADM1*	

**Table 8 cancers-14-03525-t008:** The Gene Ontology (GO) analysis and details about the top 14 upregulated and five downregulated DEGs in the control IBD enteroids vs. the control tumor enteroids. The DEGs annotated for the biological process, molecular function, and cellular component categories are shown. The percentages represent the number of genes in each functional category.

**Details about DEGs in Control IBD Enteroids vs. Control Tumor Enteroids**		
**Gene Description**	**Gene Symbol**	**Probe Set ID**
Sulfotransferase family 1C member 3	*SULT1C3*	CfaAffx.4011.1.S1_at
Osteopontin	*SPP1*	Cfa.9240.1.S1_at
Carbonic anhydrase 1	*CA1*	CfaAffx.13785.1.S1_at/Cfa.6413.1.A1_at
Mucin-1	*MUC1*	CfaAffx.26061.1.S1_at/Cfa.7074.1.A1_at
Phospholipase C epsilon 1	*PLCE1*	Cfa.12506.3.S1_at
Protein S100-A16	*S100A16*	Cfa.18759.1.S1_s_at/Cfa.18759.2.S1_at
Beta-site APP-cleaving enzyme 2 (memapsin 1) (beta-secretase 2)	*BACE2*	Cfa.20396.1.S1_at
Cell adhesion molecule 1	*CADM1*	Cfa.11274.1.A1_at/Cfa.10739.1.A1_s_at
WAP four-disulfide core domain protein 2	*WFDC2 (CE4)*	CfaAffx.15227.1.S1_s_at/Cfa.3780.1.S1_at
RNA-binding motif, single-stranded-interacting protein 1	*RBMS1*	Cfa.5781.1.A1_s_at
ENSCAFT00000022799 XM_014110988.1 PREDICTED: Canis lupus familiaris fibronectin 1 (FN1)	*FN1*	CfaAffx.22155.1.S1_s_at
Double-headed protease inhibitor, submandibular gland	*LOC111092171*	Cfa.12226.1.A1_at
Serine protease 23	*PRSS23*	Cfa.17835.1.S1_s_at
Peripheral myelin protein 22	*PMP22*	CfaAffx.27421.1.S1_at
Neutral and basic amino acid transport protein rbat	*SLC3A1*	Cfa.3561.1.S1_at
Bile acid-CoA:amino acid N-acyltransferase-like	*BAAT*	CfaAffx.4748.1.S1_at
Aminopeptidase N	*ANPEP*	Cfa.20798.1.S1_at/Cfa.3774.1.A1_s_at
Apolipoprotein A-IV	*APOA4*	Cfa.6294.1.A1_at
Annexin A13	*ANXA13*	Cfa.3796.1.A1_s_at
**Gene Ontology (GO) Analysis of DEGs (Annotated for Biological Process)**		
**GO-Terms**	**Genes**	
Primary metabolic process, organic substance metabolic process (10%)	*PRSS23*, *anpep*, *BAAT*, *SLC3A1*, *LOC111092171*, *APOA4*, *BACE2*, *WFDC2 (CE4)*	
Nitrogen compound metabolic process (9%)	*PRSS23*, *ANPEP*, *BAAT*, *LOC111092171*, *APOA4*, *BACE2*, *WFDC2 (CE4)*	
Cellular metabolic process (9%)	*CA1*, *ANPEP*, *BAAT*, *LOC111092171*, *APOA4*, *BACE2*, *WFDC2* (*CE4*)	
Regulation of cellular process (9%)	*ANXA13*, *ANPEP*, *PLCE1*, *CADM1*, *LOC111092171*, *APOA4*, *WFDC2* (*CE4*)	
Small molecule metabolic process (5%)	*CA1*, *BAAT*, *APOA4*, *BACE2*	
Regulation of metabolic process (4%)	*LOC111092171*, *APOA4*, *WFDC2 (CE4)*	
Negative regulation of cellular process (4%)	*ANXA13*, *LOC111092171*, *WFDC2 (CE4)*	
Regulation of molecular function, hydrolase activity (4%)	*LOC111092171*, *APOA4*, *WFDC2 (CE4)*	
Cell communication, signal transduction, cellular response to stimulus (4%)	*ANPEP*, *PLCE1*, *CADM1*	
Biosynthetic process (4%)	*BAAT*, *APOA4*, *BACE2*	
Negative regulation of metabolic process (2%)	*LOC111092171*, *WFDC2 (CE4)*	
Negative regulation of catalytic activity (2%)	*LOC111092171*, *WFDC2 (CE4)*	
Response to stress (2%)	*APOA4*, *WFDC2 (CE4)*	
Positive regulation of cellular process, regulation and establishment of localization/macromolecule localization (2%)	*ANXA13*, *APOA4*	
Cellular component organization or biogenesis (2%)	*APOA4*, *PMP22*	
System process, catabolic process, regulation of biological quality (2%)	*ANPEP*, *APOA4*	
Response to chemical (2%)	*S100A16*, *APOA4*	
Multicellular organism development, cellular developmental process, anatomical structure development/morphogenesis (2%)	*ANPEP*, *PMP22*	
Cell death, apoptotic process (2%)	*PMP22*, *CADM1*	
Cell adhesion (2%)	*SPP1*, *CADM1*	
Immune response, response to biotic stimulus/other organism/external stimulus (1%)	*WFDC2 (CE4)*	
Heart development, glomerulus development, regulation of smooth muscle contraction, Regulation of signaling, cell growth, cytoskeleton organization (1%)	*PLCE1*	
Lipid catabolic process, diacylglycerol biosynthetic process (1%)	*PLCE1*	
Ensheathment of neurons (1%)	*PMP22*	
Biological process involved in symbiotic interaction (1%)	*ANPEP*	
Ossification (1%)	*SPP1*	
Cellular localization, positive/negative regulation of establishment of protein localization (1%)	*ANXA13*	
Positive/negative regulation of transport, vesicle-mediated transport (1%)	*ANXA13*	
Regulation of multicellular organismal process, digestion (1%)	*APOA4*	
Plasma lipoprotein particle organization, regulation of plasma lipoprotein particle levels, positive regulation of metabolic process/catalytic activity (1%)	*APOA4*	
Immune system process, positive regulation of cytokine production/natural killer cell mediated cytotoxicity, detection of stimulus (1%)	*CADM1*	
Liver/brain development, spermatogenesis, cell differentiation/recognition (1%)	*CADM1*	
Response to calcium ion (1%)	*S100A16*	
Amyloid-beta metabolic process, glucose homeostasis, membrane protein ectodomain proteolysis, negative regulation of amyloid precursor protein biosynthetic process, peptide hormone processing (1%)	*BACE2*	
**Gene Ontology (GO) Analysis of DEGs (Annotated for Molecular Function)**		
**GO-Terms**	**Genes**	
Ion binding (18%)	*ANXA13*, *CA1*, *ANPEP*, *S100A16*, *APOA4*	
Hydrolase activity (14%)	*CA1*, *PRSS23*, *ANPEP*, *BAAT*	
Enzyme regulator activity (11%)	*LOC111092171*, *APOA4*, *WFDC2 (CE4)*	
Catalytic activity, acting on a protein (7%)	*PRSS23*, *ANPEP*	
Organic cyclic compound binding (7%)	*RBMS1*, *APOA4*	
Transferase activity (7%)	*BAAT*, *SULT1C3*	
Lipid binding (7%)	*ANXA13*, *APOA4*	
Lyase activity (4%)	*CA1*	
Enzyme binding, metal ion binding (4%)	*PLCE1*	
Phosphatidylinositol phospholipase C activity, small GTPase binding, guanyl-nucleotide exchange factor activity (4%)	*PLCE1*	
Signaling receptor binding (4%)	*CADM1*	
Cell adhesion molecule binding, PDZ domain binding, protein homodimerization activity (4%)	*CADM1*	
Calcium ion/calcium-dependent protein binding, protein homodimerization activity (4%)	*S100A16*	
RNA binding (4%)	*S100A16*	
Aspartic-type endopeptidase activity (4%)	*BACE2*	
**Gene Ontology (GO) Analysis of DEGs (Annotated for Cellular Component)**		
**GO-Terms**	**Genes**	
Intracellular anatomical structure (14%)	*ANXA13*, *MUC1*, *CA1*, *PRSS23*, *ANPEP*, *BAAT, PLCE1, S100A16, CADM1, BACE2*	
Cytoplasm (13%)	*ANXA13*, *CA1*, *MUC1*, *ANPEP*, *BAAT*, *PLCE1*, *S100A16*, *CADM1*, *BACE2*	
Membrane/plasma membrane (13%)	*ANXA13*, *MUC1*, *PRSS23*, *ANPEP*, *S100A16*, *SLC3A1*, *PMP22*, *CADM1*, *BACE2*	
Organelle (11%)	*ANXA13*, *MUC1*, *PRSS23*, *ANPEP*, *BAAT*, *PLCE1*, *S100A16*, *BACE2*	
Extracellular region (10%)	*ANPEP*, *PLCE1*, *S100A16*, *LOC111092171*, *SPP1*, *APOA4*, *WFDC2 (CE4)*	
Intrinsic component of membrane (10%)	*MUC1*, *PRSS23*, *ANPEP*, *SLC3A1*, *CADM1*, *PMP22*, *BACE2*	
Extracellular space (7%)	*ANPEP*, *PLCE1*, *S100A16*, *APOA4*, *WFDC2 (CE4)*	
Cell periphery (7%)	*ANXA13*, *MUC1*, *ANPEP*, *S100A16*, *PMP22*	
Apical part of cell (3%)	*ANXA13*, *MUC1*	
Cytosol (3%)	*BAAT*, *S100A16*	
Membrane-enclosed lumen (3%)	*ANXA13*, *S100A16*	
Endomembrane system (3%)	*ANXA13*, *PLCE1*	
Golgi apparatus membrane, lamellipodium (1%)	*PLCE1*	
Basolateral plasma membrane, cell–cell junction, neuron projection, postsynaptic density (1%)	*CADM1*	
Nucleus, nucleolus (1%)	*S100A16*	
Endoplasmic reticulum, Golgi apparatus, endosome (1%)	*BACE2*	

**Table 9 cancers-14-03525-t009:** The Gene Ontology (GO) analysis and details about the DEGs in the LPS treated IBD colonoids vs. the LPS treated IBD enteroids annotated for the biological process, molecular function, and cellular component categories. The number of genes that fell into each of these categories is shown in percentages.

**Details about DEGs in LPS Treated IBD Colonoids vs. LPS Treated IBD Enteroids**		
**Probe Set ID**	**Gene Description**	**Gene Symbol**
CfaAffx.13785.1.S1_at/Cfa.6413.1.A1_at	Carbonic anhydrase 1	*CA1*
CfaAffx.31015.1.S1_at	Chloride channel accessory 1	*CLCA1*
CfaAffx.4011.1.S1_at	Sulfotransferase 1C1	*SULT1C1*
Cfa.7074.1.A1_at/CfaAffx.26061.1.S1_at	Mucin-1	*MUC1*
Cfa.16455.1.S1_at	C-X-C motif chemokine 13	*CXCL13*
Cfa.15827.1.S1_s_at	Fucosyltransferase 5 (alpha (1,3) fucosyltransferase)	*FUT5* (*fut356*)
Cfa.4456.1.S1_at	proSAAS	*PCSK1N*
Cfa.18759.2.S1_at	Protein S100-A16	*S100A16*
CfaAffx.8210.1.S1_at	ENSCAFT00000007801 XM_022408371.1 PREDICTED: *Canis lupus familiaris* olfactomedin 4 (OLFM4), mRNA	*OLFM4*
Cfa.5989.1.A1_s_at	Alpha-1-antitrypsin-like	*SERPINA1*
CfaAffx.3936.1.S1_at	HEPACAM family member 2	*HEPACAM2*
CfaAffx.15227.1.S1_s_at	WAP four-disulfide core domain protein 2	*WFDC2* (*CE4*)
Cfa.21325.1.S1_s_at	Epithelial membrane protein 2	*EMP2*
Cfa.12271.1.A1_at	Ras-related protein Rap-2a-like	*LOC480441*
CfaAffx.9034.1.S1_s_at	ENSCAFT00000008689 XM_022406088.1 PREDICTED *Canis lupus familiaris* Fc fragment of IgG binding protein FCGBP mRNA	*FCGBP*
Cfa.496.1.A1_s_at	Collagen alpha-2(V) chain	*COL5A2*
CfaAffx.14168.1.S1_s_at	Annexin A10	*ANXA10*
Cfa.11104.1.S1_at	Phytanoyl-CoA hydroxylase-like	*LOC478000*
Cfa.18640.1.S1_at	Long-chain-fatty-acid-CoA ligase 5	*ACSL5*
Cfa.10909.1.A1_at	39S ribosomal protein S30, mitochondrial	*MRPS30*
CfaAffx.4748.1.S1_at	Bile acid-CoA:amino acid N-acyltransferase-like	*BAAT*
Cfa.6996.1.A1_at	Ectonucleotide pyrophosphatase/phosphodiesterase family member 6	*ENPP6*
Cfa.6294.1.A1_at	Apolipoprotein A-IV	*APOA4*
**Gene Ontology (GO) Analysis of DEGs (Annotated for Biological Process)**		
**GO-Terms**	**Genes**	
Cellular metabolic process (7%)	*PCSK1N*, *CA1*, *BAAT*, *FUT5* (*fut356*), *APOA4*, *ACSL5*, *WFDC2* (*CE4*), *EMP2*	
Organic substance/nitrogen compound metabolic process (7%)	*PCSK1N*, *BAAT*, *FUT5* (*fut356*), *ENPP6*, *APOA4*, *ACSL5*, *WFDC2* (*CE4*), *EMP2*	
Regulation of cellular process (7%)	*PCSK1N*, *CLCA1*, *LOC480441*, *CXCL13*, *APOA4*, *ACSL5*, *WFDC2 (CE4)*, *EMP2*	
Primary metabolic process (7%)	*PCSK1N*, *BAAT*, *ENPP6*, *FUT5 (fut356)*, *APOA4*, *ACSL5*, *WFDC2 (CE4)*, *EMP2*	
Regulation of molecular function (5%)	*PCSK1N*, *CLCA1*, *LOC480441*, *APOA4*, *WFDC2 (CE4)*, *EMP2*	
Positive regulation of cellular process (4%)	*CLCA1*, *CXCL13*, *APOA4*, *ACSL5*, *EMP2*	
Small molecule metabolic process (3%)	*CA1*, *BAAT*, *APOA4*, *ACSL5*	
Regulation of metabolic process (3%)	*PCSK1N*, *APOA4*, *WFDC2 (CE4)*, *EMP2*	
Signal transduction, cell communication, cellular response to stimulus (3%)	*CLCA1*, *LOC480441*, *CXCL13*, *EMP2*	
Response to stress (3%)	*CLCA1*, *CXCL13*, *APOA4*, *WFDC2 (CE4)*	
Biosynthetic process (3%)	*BAAT*, *FUT5 (fut356)*, *APOA4*, *ACSL5*	
Regulation of hydrolase activity (3%)	*PCSK1N*, *CLCA1*, *APOA4*, *WFDC2 (CE4)*	
Regulation of localization (3%)	*CXCL13*, *APOA4*, *ACSL5*, *EMP2*	
Immune response (3%)	*CXCL13*, *WFDC2 (CE4)*, *EMP2*	
Macromolecule localization and establishment (3%)	*APOA4*, *ACSL5*, *EMP2*	
Regulation of biological quality/multicellular organismal process (3%)	*CXCL13*, *APOA4*, *EMP2*	
Negative regulation of cellular process (3%)	*PCSK1N*, *CXCL13*, *WFDC2 (CE4)*	
Response to other organism/external stimulus/biotic stimulus (3%)	*CLCA1*, *CXCL13*, *WFDC2 (CE4)*	
Positive regulation of catalytic activity (2%)	*APOA4*, *EMP2*	
Movement of cell or subcellular component, cell motility, localization of cell, positive regulation of response to stimulus (2%)	*CXCL13*, *EMP2*	
Response to chemical (2%)	*CXCL13*, *APOA4*	
Negative regulation of catalytic activity (2%)	*PCSK1N*, *WFDC2 (CE4)*	
Taxis, leukocyte migration (2%)	*CLCA1*, *CXCL13*	
Regulation of signaling (2%)	*CLCA1*, *EMP2*	
System process, cellular component organization or biogenesis, positive regulation of metabolic process (2%)	*APOA4*, *EMP2*	
Catabolic process (2%)	*ENPP6*, *APOA4*	
Regulation of locomotion, tissue migration (2%)	*CXCL13*, *EMP2*	
Negative regulation of metabolic process (2%)	*PCSK1N*, *WFDC2 (CE4)*	
Positive regulation of signaling (1%)	*EMP2*	
Transmembrane transport, positive regulation of transport/lipid localization (1%)	*ACSL5*	
Digestion, plasma lipoprotein particle organization, regulation of plasma lipoprotein particle levels (1%)	*APOA4*	
Cell–cell signaling, cellular homeostasis, response to endogenous stimulus, positive regulation of immune system process/locomotion, negative regulation of locomotion/multicellular organismal process/response to stimulus (1%)	*CXCL13*	
Cell population proliferation, actin filament-based process, cell adhesion, cell death/cell killing, multi-multicellular organism process/development, anatomical structure formation involved in morphogenesis, cellular developmental process, cellular localization, vesicle-mediated transport, embryo implantation, multi-organism reproductive process, immune effector process, regulation of developmental process/transferase activity (1%)	*EMP2*	
Protein glycosylation (1%)	*FUT5 (fut356)*	
Calcium ion transport, cellular response to hypoxia, chloride transport, ion transmembrane transport (1%)	*CLCA1*	
Actin cytoskeleton reorganization, cellular response to xenobiotic stimulus, Rap protein signal transduction, regulation of dendrite morphogenesis, JNK cascade (1%)	*LOC480441*	
Establishment of protein localization, protein localization to plasma membrane, microvillus assembly, negative regulation of cell migration (1%)	*LOC480441*	
Positive regulation of protein phosphorylation/autophosphorylation (1%)	*LOC480441*	
Methyl-branched fatty acid/2-oxoglutarate/isoprenoid metabolic process, 2-oxobutyrate catabolic process (1%)	*LOC478000*	
Oxidation-reduction process, fatty acid alpha-oxidation (1%)	*LOC478000*	
Cell division, centrosome cycle (1%)	*HEPACAM2*	
**Gene Ontology (GO) Analysis of DEGs (Annotated for Molecular Function)**		
**GO-Terms**	**Genes**	
Cation binding (17%)	*CA1*, *ANXA10*, *S100A16*, *COL5A2*, *APOA4*	
Hydrolase activity, acting on ester bonds (10%)	*CA1*, *BAAT*, *ENPP6*	
Signaling receptor binding (7%)	*CXCL13*, *EMP2*	
Enzyme inhibitor activity (7%)	*PCSK1N*, *WFDC2 (CE4)*	
Phospholipid binding (7%)	*ANXA10*, *APOA4*	
Peptidase regulator activity (7%)	*PCSK1N*, *WFDC2 (CE4)*	
Hydrolase activity, acting on acid anhydrides, ligase activity, forming carbon-sulfur bonds (3%)	*ACSL5*	
Phosphatidylcholine binding, enzyme activator activity, alcohol binding, steroid binding (3%)	*APOA4*	
Transferase activity, transferring acyl groups (3%)	*BAAT*	
Carbon-oxygen lyase activity (3%)	*CA1*	
Glycosaminoglycan binding, heparin binding, growth factor binding (3%)	*CXCL13*	
Enzyme binding, cell adhesion molecule binding, integrin binding (3%)	*EMP2*	
Transferase activity, transferring glycosyl groups, fucosyltransferase activity (3%)	*FUT5 (fut356)*	
Transferase activity, transferring sulfur-containing groups (3%)	*SULT1C3*	
Intracellular calcium activated chloride channel activity, metal ion binding, metalloendopeptidase activity (3%)	*CLCA1*	
Signaling receptor activator activity (3%)	*CXCL13*	
GDP/GTP binding, G protein activity, GTPase activity, magnesium ion binding (3%)	*LOC480441*	
Protein/L-ascorbic acid/carboxylic acid/ferrous iron binding (3%)	*LOC478000*	
Phytanoyl-CoA dioxygenase activity (3%)	*LOC478000*	
**Gene Ontology (GO) Analysis of DEGs (Annotated for Cellular Component)**		
**GO-Terms**	**Genes**	
Intracellular organelle (14%)	*MUC1*, *BAAT*, *FUT5 (fut356)*, *ACSL5*, *EMP2*, *HEPACAM2*	
Integral component of membrane (14%)	*MUC1*, *FUT5 (fut356)*, *HEPACAM2*, *ACSL5*, *EMP2*	
Membrane-bounded organelle (14%)	*MUC1*, *BAAT*, *FUT5 (fut356)*, *ACSL5*, *EMP2*, *HEPACAM2*	
Plasma membrane (14%)	*MUC1*, *ENPP6*, *ACSL5*, *EMP2*, *CLCA1*, *LOC480441*	
Plasma membrane region, apical plasma membrane (5%)	*MUC1*, *EMP2*	
Organelle membrane, Golgi apparatus subcompartment (5%)	*FUT5 (fut356)*, *EMP2*	
Mitochondrion (5%)	*ACSL5*, *LOC478000*	
Plasma lipoprotein particle (3%)	*APOA4*	
Intracellular non-membrane-bounded organelle, intracellular organelle lumen/Endoplasmic reticulum (3%)	*ACSL5*	
Lipoprotein particle (3%)	*APOA4*	
Membrane microdomain, intracellular vesicle/cytoplasmic vesicle (3%)	*EMP2*	
Microbody (3%)	*BAAT*	
Integral component of plasma membrane (3%)	*CLCA1*	
Extracellular space, microvillus, zymogen granule membrane (3%)	*CLCA1*	
Golgi cisterna membrane (3%)	*FUT5 (fut356)*	
Cytosol, recycling endosome membrane, midbody (3%)	*LOC480441*	
Peroxisome, 9 + 0 non-motile cilium (3%)	*LOC478000*	
Cytoskeleton, centrosome, mitotic spindle, Golgi membrane, Nucleus, nucleoplasm, midbody (3%)	*HEPACAM2*	

**Table 10 cancers-14-03525-t010:** The Gene Ontology (GO) analysis and details about the top 17 upregulated and six downregulated DEGs in the LPS treated IBD enteroids vs. the LPS treated tumor enteroids. The DEGs annotated for the biological process, molecular function, and cellular component categories are shown. The percentages represent the number of genes in each functional category.

**Details about DEGs in LPS Treated IBD Enteroids vs. LPS Treated Tumor Enteroids**		
**Probe Set ID**	**Gene Description**	**Gene Symbol**
CfaAffx.13785.1.S1_at/Cfa.6413.1.A1_at	Carbonic anhydrase 1	*CA1*
CfaAffx.31015.1.S1_at	Chloride channel accessory 1	*CLCA1*
CfaAffx.4011.1.S1_at	Sulfotransferase family 1C member 3	*SULT1C3*
Cfa.7074.1.A1_at	Mucin-1	*MUC1*
Cfa.16455.1.S1_at	C-X-C motif chemokine 13	*CXCL13*
Cfa.15827.1.S1_s_at	Fucosyltransferase 5 (alpha (1,3) fucosyltransferase)	*FUT5 (fut356)*
Cfa.4456.1.S1_at	proSAAS	*PCSK1N*
Cfa.496.1.A1_s_at	Collagen alpha-2(V) chain	*COL5A2*
CfaAffx.15227.1.S1_s_at/Cfa.3780.1.S1_at	WAP four-disulfide core domain protein 2	*WFDC2 (CE4)*
CfaAffx.14168.1.S1_s_at	Annexin A10	*ANXA10*
CfaAffx.3936.1.S1_at	HEPACAM family member 2 isoform X1	*HEPACAM2*
Cfa.5989.1.A1_s_at	Alpha-1-antitrypsin-like	*SERPINA1*
CfaAffx.16198.1.S1_s_at	Trefoil factor 1	*TFF1*
CfaAffx.9034.1.S1_s_at	ENSCAFT00000008689 XM_022406088.1 PREDICTED: Canis lupus familiaris Fc fragment of IgG binding protein (FCGBP)	*FCGBP*
Cfa.12271.1.A1_at	Ras-related protein Rap-2a-like	*LOC480441*
Cfa.21325.1.S1_s_at	Epithelial membrane protein 2	*EMP2*
CfaAffx.8210.1.S1_at	ENSCAFT00000007801 XM_022408371.1 PREDICTED: Canis lupus familiaris olfactomedin 4 (OLFM4)	*OLFM4*
Cfa.10909.1.A1_at	39S Ribosomal protein S30, mitochondrial	*MRPS30*
Cfa.15806.1.S1_at	C-C motif chemokine 25	*CCL25*
Cfa.18640.1.S1_at	Long-chain-fatty-acid--CoA ligase 5	*ACSL5*
CfaAffx.4748.1.S1_at	Bile acid-CoA:amino acid N-acyltransferase-like	*BAAT*
Cfa.6996.1.A1_at	Ectonucleotide pyrophosphatase/phosphodiesterase family member 6	*ENPP6*
Cfa.6294.1.A1_at	Apolipoprotein A-IV	*APOA4*
**Gene Ontology (GO) Analysis of DEGs (Annotated for Biological Process)**		
**GO-Terms**	**Genes**	
Regulation of cellular process (7%)	*PCSK1N*, *CCL25*, *CLCA1*, *LOC480441*, *CXCL13*, *TFF1*, *APOA4*, *ACSL5*, *WFDC2 (CE4)*, *EMP2*	
Cellular metabolic process (6%)	*PCSK1N*, *CA1*, *BAAT*, *FUT5 (fut356)*, *APOA4*, *ACSL5*, *WFDC2 (CE4)*, *EMP2*	
Primary metabolic process, organic substance/nitrogen compound metabolic process (6%)	*PCSK1N*, *BAAT*, *FUT5 (fut356)*, *ENPP6*, *ACSL5*, *APOA4*, *WFDC2 (CE4)*, *EMP2*	
Regulation of molecular function (5%)	*PCSK1N*, *CCL25*, *CLCA1*, *LOC480441*, *APOA4*, *WFDC2 (CE4)*, *EMP2*	
Cell communication, signal transduction, cellular response to stimulus (4%)	*CCL25*, *CLCA1*, *LOC480441*, *CXCL13*, *TFF1*, *EMP2*	
Positive regulation of cellular process (4%)	*CCL25*, *CLCA1*, *CXCL13*, *APOA4*, *ACSL5*, *EMP2*	
Response to stress (3%)	*CCL25*, *CLCA1*, *CXCL13*, *APOA4*, *WFDC2 (CE4)*	
Regulation of hydrolase activity (3%)	*PCSK1N*, *CCL25*, *CLCA1*, *APOA4*, *WFDC2 (CE4)*	
Immune response (3%)	*CCL25*, *CXCL13*, *WFDC2 (CE4)*, *EMP2*	
Regulation of metabolic process (3%)	*PCSK1N*, *APOA4*, *WFDC2 (CE4)*, *EMP2*	
Small molecule metabolic process (3%)	*CA1*, *BAAT*, *APOA4*, *ACSL5*	
Response to biotic stimulus/external stimulus/other organism (3%)	*CCL25*, *CLCA1*, *CXCL13*, *WFDC2 (CE4)*	
Regulation of biological quality (3%)	*CXCL13*, *TFF1*, *APOA4*, *EMP2*	
Cell motility, movement of cell or subcellular component, positive regulation of response to stimulus, localization of cell (3%)	*CCL25*, *CLCA1*, *CXCL13*, *EMP2*	
Biosynthetic process (3%)	*BAAT*, *FUT5 (fut356)*, *APOA4*, *ACSL5*	
Response to chemical (3%)	*CCL25*, *CLCA1*, *CXCL13*, *APOA4*	
Regulation of localization (3%)	*CXCL13*, *APOA4*, *ACSL5*, *EMP2*	
Negative regulation of cellular process (2%)	*PCSK1N*, *CXCL13*, *WFDC2 (CE4)*	
Positive regulation of catalytic activity (2%)	*CCL25*, *APOA4*, *EMP2*	
Negative regulation of catalytic activity (2%)	*PCSK1N*, *LOC480441*, *WFDC2 (CE4)*	
Macromolecule localization (2%)	*APOA4*, *ACSL5*, *EMP2*	
Leukocyte migration, taxis (2%)	*CCL25*, *CLCA1*, *CXCL13*	
Regulation of signaling (2%)	*CCL25*, *CLCA1*, *EMP2*	
Establishment of localization (2%)	*APOA4*, *ACSL5*, *EMP2*	
Regulation of multicellular organismal process (2%)	*CXCL13*, *APOA4*, *EMP2*	
System process (2%)	*TFF1*, *APOA4*, *EMP2*	
Positive regulation of signaling (2%)	*CCL25*, *CLCA1*, *EMP2*	
Negative regulation of metabolic process (1%)	*PCSK1N*, *WFDC2 (CE4)*	
Cellular component organization or biogenesis (1%)	*APOA4*, *EMP2*	
Catabolic process (1%)	*ENPP6*, *APOA4*	
Tissue migration (1%)	*CXCL13*, *EMP2*	
Digestion (1%)	*TFF1*, *APOA4*	
Positive regulation of metabolic process (1%)	*APOA4*, *EMP2*	
Regulation of locomotion (1%)	*CXCL13*, *EMP2*	
Transmembrane transport, positive regulation of transport/lipid localization (1%)	*ACSL5*	
Regulation of plasma lipoprotein particle levels/organization (1%)	*APOA4*	
Cell–cell signaling, negative regulation of response to stimulus/multicellular organismal process, regulation of immune system process, pos. regulation of immune system process, response to endogenous stimulus, cellular homeostasis, pos./neg. regul. of locomotion (1%)	*CXCL13*	
Multi-multicellular organism process, multicellular organism development, cellular developmental process and regulation, anatomical structure development/formation involved in morphogenesis, cellular localization, cell population proliferation, actin filament-based process, immune effector process, multi-organism reproductive process, cell adhesion, regulation of transferase activity, cell death/cell killing, embryo implantation, vesicle-mediated transport (1%)	*EMP2*	
Glycosylation (1%)	*FUT5 (fut356)*	
Multicellular organismal homeostasis (1%)	*TFF1*	
Calcium ion transport, cellular response to hypoxia, chloride transport, ion transmembrane transport (1%)	*CLCA1*	
Actin cytoskeleton reorganization, cellular response to xenobiotic stimulus, Rap protein signal transduction, regulation of dendrite morphogenesis, JNK cascade (1%)	*LOC480441*	
Establishment of protein localization, protein localization to plasma membrane, microvillus assembly, negative regulation of cell migration (1%)	*LOC480441*	
Positive regulation of protein phosphorylation/autophosphorylation (1%)	*LOC480441*	
**Gene Ontology (GO) Analysis of DEGs (Annotated for Molecular Function)**		
**GO-Terms**	**Genes**	
Receptor ligand activity (14%)	*CCL25*, *CXCL13*, *TFF1*	
Metal ion binding (14%)	*CA1*, *ANXA10*, *COL5A2*	
Cytokine receptor binding, G protein-coupled receptor binding (9%)	*CCL25*, *CXCL13*	
Peptidase inhibitor activity, endopeptidase regulator activity (9%)	*PCSK1N*, *WFDC2 (CE4)*	
Hydro-lyase activity, carboxylic ester hydrolase activity (5%)	*CA1*	
Acid-thiol ligase activity, hydrolase activity, acting on acid anhydrides, in phosphorus-containing anhydrides, CoA-ligase activity (5%)	*ACSL5*	
Calcium-dependent phospholipid binding (5%)	*ANXA10*	
Phosphatidylcholine-sterol O-acyltransferase activator activity, sterol binding, cholesterol binding (5%)	*APOA4*	
Thiolester hydrolase activity, transferase activity, transferring acyl groups other than amino-acyl groups (5%)	*BAAT*	
Fibroblast growth factor binding (5%)	*CXCL13*	
Kinase binding (5%)	*EMP2*	
Phosphoric ester hydrolase activity (5%)	*ENPP6*	
Transferase activity, transferring hexosyl/glycosyl groups (5%)	*FUT5 (fut356)*	
Sulfotransferase activity (5%)	*SULT1C3*	
Intracellular calcium activated chloride channel activity, metal ion binding, metalloendopeptidase activity (5%)	*CLCA1*	
GDP/GTP binding, G protein activity, GTPase activity, magnesium ion binding (5%)	*LOC480441*	
**Gene Ontology (GO) Analysis of DEGs (Annotated for Cellular Component)**		
**GO-Terms**	**Genes**	
Plasma membrane (23%)	*MUC1*, *ENPP6*, *CLCA1*, *ACSL5*, *EMP2*, *LOC480441*	
Intracellular organelle, membrane-bounded organelle (15%)	*BAAT*, *FUT5 (fut356)*, *ACSL5*, *EMP2*	
Integral component of membrane (15%)	*FUT5 (fut356)*, *HEPACAM2*, *ACSL5*, *EMP2*	
Organelle membrane, Golgi apparatus subcompartment (8%)	*FUT5 (fut356)*, *EMP2*	
Apical plasma membrane (8%)	*MUC1*, *EMP2*	
Endoplasmic reticulum, intracellular organelle lumen, mitochondrion, intracellular non-membrane-bounded organelle (4%)	*ACSL5*	
Plasma lipoprotein particle (4%)	*APOA4*	
Microbody (4%)	*BAAT*	
Intracellular vesicle, membrane microdomain, cytoplasmic vesicle (4%)	*EMP2*	
Integral component of plasma membrane (4%)	*CLCA1*	
Extracellular space, microvillus, zymogen granule membrane (4%)	*CLCA1*	
Golgi cisterna membrane (4%)	*FUT5 (fut356)*	
Cytosol, recycling endosome membrane, midbody (4%)	*LOC480441*	

**Table 11 cancers-14-03525-t011:** The Gene Ontology (GO) analysis and details about the DEGs exhibiting a similar pattern of expression between the IBD enteroids and colonoids following LPS stimulation annotated for the biological process, molecular function, and cellular component categories. The number of genes that fell into each of these categories is shown in percentages.

Details about 25 DEGs Exhibiting Same Trend of Expression between IBD Enteroids and Colonoids Following LPS Stimulation				
Probe Set ID	Gene Description	Gene Symbol	LPS Treated vs. Control IBD Colonoids (Contrast 3) (log2 M)	LPS Treated vs. Control IBD Enteroids (Contrast 2) (log2 M)
CfaAffx.16440.1.S1_at	2′-5′-Oligoadenylate synthase-like protein	*OASL*	0.851	0.770
CfaAffx.2163.1.S1_at	Antigen peptide transporter 2	*TAP2*	0.649	0.619
CfaAffx.16198.1.S1_s_at	Trefoil factor 1	*TFF1*	0.924	0.599
Cfa.12501.1.A1_at/CfaAffx.19068.1.S1_at/ CfaAffx.19068.1.S1_s_at	Insulin-like growth factor-binding protein 1	*IGFBP1*	0.453	0.397
CfaAffx.29762.1.S1_at	NADPH oxidase organizer 1	*NOXO1*	0.556	0.417
CfaAffx.15121.1.S1_at	Interferon-induced protein with tetratricopeptide repeats 1	*IFIT1*	0.855	0.407
CfaAffx.21951.1.S1_s_at	Eukaryotic translation elongation factor 1 alpha 1	*EEF1A1*	0.285	0.332
Cfa.21191.2.S1_a_at	2′-5′-Oligoadenylate synthase 1	*OAS1*	1.028	0.284
Cfa.10757.1.S1_at	Ubiquitin-like protein ISG15	*ISG15*	0.953	0.284
Cfa.2878.1.A1_s_at	Ceruloplasmin	*CP*	0.965	0.271
CfaAffx.14226.1.S1_s_at	DExD/H-box helicase 60	*DDX60*	0.515	0.267
CfaAffx.17868.1.S1_at	Glutathione peroxidase 1	*GPX1*	0.343	0.213
CfaAffx.18742.1.S1_at	Eukaryotic translation initiation factor 3 subunit A	*EIF3A*	0.306	0.185
Cfa.20996.1.S1_at	MHC class I DLA-12	*DLA-12*	0.348	0.118
Cfa.15976.1.S1_at	ATP synthase F1 subunit epsilon	*ATP5F1E*	−0.282	−0.085
CfaAffx.23922.1.S1_x_at	40S ribosomal protein S24	*RPS24*	−0.641	−0.155
Cfa.5563.1.A1_s_at	Protein disulfide isomerase family A member 6	*PDIA6*	−0.332	−0.180
Cfa.3039.1.A1_at	ATPase Na+/K+ transporting subunit alpha 1	*ATP1A1*	−0.158	−0.189
Cfa.9039.1.A1_at	Fos proto-oncogene, AP-1 transcription factor subunit	*FOS*	−0.427	−0.190
Cfa.1408.1.S1_at	Reticulon 4	*RTN4*	−0.226	−0.303
Cfa.12512.1.A1_at	Ecm29 proteasome adaptor and scaffold	*ECPAS*	−0.196	−0.387
Cfa.31.1.S1_s_at	Acidic nuclear phosphoprotein 32 family member A	*ANP32A*	−0.428	−0.396
Cfa.15462.1.A1_at	Metallothionein-1	*LOC100686073*	−0.522	−0.489
Cfa.7907.2.A1_a_at	Uncharacterized LOC611209	*LOC611209*	−0.530	−0.645
CfaAffx.11648.1.S1_at	Chromodomain helicase DNA binding protein 7	*CHD7*	−0.870	−0.834
**Gene Ontology (GO) Analysis of DEGs (Annotated for Biological Process)**				
**GO-Terms**		**Genes**		
Response to external biotic stimulus, defense response to other organism/virus, viral process, innate immune response, negative regulation of viral genome replication (6%)		*OAS1*, *OASL*, *IFIT1*, *ISG15*		
Homeostatic process (6%)		*CP*, *TFF1*, *IFIT1*, *ISG15*		
Macromolecule metabolic process (6%)		*OASL*, *EEF1A1*, *RPS24*, *ISG15*		
Cellular nitrogen compound metabolic process (6%)		*OAS1*, *OASL*, *EEF1A1*, *RPS24*		
Organonitrogen compound metabolic process (6%)		*OAS1*, *EEF1A1*, *RPS24*, *ISG15*		
Transport (6%)		*TAP2*, *CP*, *ATP1A1*, *IFIT1*		
Regulation of viral process (6%)		*OAS1*, *OASL*, *IFIT1*, *ISG15*		
Cellular macromolecule/protein metabolic process (4%)		*EEF1A1*, *RPS24*, *ISG15*		
Cellular/organic substance biosynthetic process (4%)		*OAS1*, *EEF1A1*, *RPS24*		
Response to organic substance (3%)		*IFIT1*, *ISG15*		
Regulation of cellular/primary/macromolecule/nitrogen compound metabolic process (3%)		*OASL*, *ISG15*		
Cellular aromatic/organic cyclic/nucleobase-containing/heterocycle compound metabolic process (3%)		*OAS1*, *OASL*		
Cell surface receptor signaling pathway (3%)		*IGFBP1*, *ISG15*		
Cellular response to chemical stimulus (3%)		*IFIT1*, *GPX1*		
Regulation of transport, ion transmembrane transport (1%)		*ATP1A1*		
Cellular chemical homeostasis (1%)		*CP*		
Negative regulation of ERK1 and ERK2 cascade, formation of cytoplasmic translation initiation complex, IRES-dependent viral translational initiation, translational initiation, translation reinitiation, viral translational termination–reinitiation (1%)		*EIF3A*		
Response to toxic substance/oxidative stress, cellular oxidant detoxification (1%)		*GPX1*		
Response to nitrogen compound, establishment of protein localization (1%)		*IFIT1*		
Positive regulation of viral genome replication (1%)		*IFIT1*		
Multi-organism transport, protein localization, biological process involved in interaction with host (1%)		*IFIT1*		
Negative regulation of molecular function/protein binding/hydrolase activity (1%)		*IFIT1*		
Regulation of ATPase activity/immune effector process (1%)		*IFIT1*		
Regulation of response to biotic stimulus/stress/external stimulus (1%)		*IFIT1*		
Regulation of cell communication/signal transduction (1%)		*IGFBP1*		
Response to bacterium (1%)		*ISG15*		
Cellular/organic substance catabolic process, negative regulation of cellular/macromolecule/nitrogen compound metabolic process (1%)		*ISG15*		
Interferon-gamma/Interleukin-10 production, regulation of cytokine production (1%)		*ISG15*		
Animal organ/system/tissue development, erythrocyte homeostasis, hemopoiesis, hematopoietic or lymphoid organ development, regulation of multicellular organismal/biomineral tissue development (1%)		*ISG15*		
Bone mineralization, positive regulation of cell differentiation, ossification, biomineral tissue development (1%)		*ISG15*		
Reactive oxygen species metabolic process (1%)		*NOXO1*		
Positive regulation of molecular function (1%)		*NOXO1*		
Cellular component organization (1%)		*NOXO1*		
Organophosphate/phosphorus/nucleobase-containing small molecule metabolic process (1%)		*OAS1*		
Regulation of nuclease activity (1%)		*OASL*		
Adaptive immune response, antigen processing and presentation of endogenous/peptide antigen (1%)		*TAP2*		
Tissue homeostasis, digestive system process (1%)		*TFF1*		
Immune response, antigen processing and presentation (1%)		*DLA-12*		
Cellular zinc ion homeostasis, cellular response to cadmium ion/chromate/copper ion/zinc ion, detoxification of copper ion (1%)		*LOC100686073*		
Negative regulation of growth/neuron apoptotic process, nitric oxide mediated signal transduction (1%)		*LOC100686073*		
**Gene Ontology (GO) Analysis of DEGs (Annotated for Molecular Function)**				
**GO-Terms**		**Genes**		
Nucleic acid binding (16%)		*OAS1*, *OASL*, *EEF1A1*, *IFIT1*, *EIF3A*		
Ribonucleotide binding, nucleoside phosphate binding, anion binding (9%)		*TAP2*, *EEF1A1*, *ATP1A1*		
Signaling receptor binding (9%)		*TAP2*, *TFF1*, *ISG15*		
Cation binding (6%)		*CP*, *ATP1A1*		
Enzyme binding (6%)		*NOXO1*, *ISG15*		
Active/ion transmembrane transporter activity (6%)		*TAP2*, *ATP1A1*		
Transferase activity, transferring phosphorus-containing groups (6%)		*OAS1*, *OASL*		
Hydrolase activity, acting on acid anhydrides (6%)		*TAP2*, *EEF1A1*		
Inorganic molecular entity transmembrane transporter activity (3%)		*ATP1A1*		
Oxidoreductase activity, oxidizing metal ions (3%)		*CP*		
Nucleoside binding, translation factor activity, RNA binding (3%)		*EEF1A1*		
mRNA binding, receptor tyrosine kinase binding, structural molecule activity, translation initiation factor activity (3%)		*EIF3A*		
Oxidoreductase activity, glutathione peroxidase activity, acting on peroxide as acceptor (3%)		*GPX1*		
Growth factor binding (3%)		*IGFBP1*		
Integrin binding, cell adhesion molecule binding (3%)		*ISG15*		
Enzyme activator activity, phospholipid binding (3%)		*NOXO1*		
Amide transmembrane transporter activity (3%)		*TAP2*		
Signaling receptor activator activity (3%)		*TFF1*		
Metal ion (copper ion/zinc ion) binding (3%)		*LOC100686073*		
**Gene Ontology (GO) Analysis of DEGs (Annotated for Cellular Component)**				
**GO-Terms**		**Genes**		
Intracellular organelle (15%)		*TAP2*, *OASL*, *RPS24*, *IGFBP1*, *ISG15*		
Integral component of membrane (12%)		*TAP2*, *ATP1A1*, *NOXO1*, *DLA-12*		
Membrane-bounded organelle (12%)		*TAP2*, *OASL*, *IGFBP1*, *ISG15*		
Intracellular non-membrane-bounded organelle (9%)		*OASL*, *RPS24*, *ISG15*		
Plasma membrane (6%)		*ATP1A1*, *NOXO1*		
Endoplasmic reticulum (6%)		*DLA-12*, *TAP2*		
Nucleus, cytosol, cytoplasm (6%)		*EIF3A*, *LOC100686073*		
Host cell (3%)		*IFIT1*		
Golgi apparatus (3%)		*IGFBP1*		
Cytosolic ribosome, ribosomal subunit (3%)		*ISG15*		
Intrinsic component of plasma membrane, plasma membrane protein complex (3%)		*NOXO1*		
Oxidoreductase complex (3%)		*NOXO1*		
Intracellular organelle lumen (3%)		*OASL*		
TAP complex, MHC class I peptide loading complex (3%)		*TAP2*		
Endoplasmic reticulum subcompartment, nuclear outer membrane-endoplasmic reticulum membrane network, intrinsic component of organelle membrane (3%)		*TAP2*		
Integral component of lumenal side of endoplasmic reticulum membrane, phagocytic vesicle membrane (3%)		*DLA-12*		
Lysosome (3%)		*LOC100686073*		
Membrane, cytoskeleton, microtubule, nucleolus, nucleoplasm, postsynaptic density, multi-eIF complex, eukaryotic 43S/48S preinitiation complex, eukaryotic translation initiation factor 3 complex, eIF3e/eIF3m (3%)		*EIF3A*		

**Table 12 cancers-14-03525-t012:** A list of genes in the common/same direction between the IBD enteroids, colonoids, and tumor-adjacent organoids following LPS stimulation. The log-ratio *M* values represent the log(R/G) (log fold change) [44].

Probe Set ID	Gene Description	Gene Symbol	LPS Treated vs. Control IBD Colonoids (Log-Ratio *M*)	LPS Treated vs. Control IBD Enteroids (Log-Ratio *M*)	LPS Treated vs. Control Tumor-Adjacent Organoids (Log-Ratio *M*)
CfaAffx.14226.1.S1_s_at	DEAD (Asp-Glu-Ala-Asp) box polypeptide 60	*DDX60*	0.515	0.267	0.499
CfaAffx.18742.1.S1_at	Eukaryotic translation initiation factor 3 subunit A	*EIF3A*	0.306	0.185	0.215
Cfa.15462.1.A1_at	Metallothionein-1	*LOC100686073*	−0.522	−0.489	−0.556

**Table 13 cancers-14-03525-t013:** The Gene Ontology (GO) analysis and the details about the 20 DEGs exhibiting an opposite trend (upregulated in IBD enteroids and downregulated in the IBD colonoids and vice versa) between the IBD enteroids and colonoids following LPS stimulation annotated for the biological process, molecular function, and cellular component categories. The percentages represent the number of genes in each functional category.

Details about 20 DEGs Exhibiting Opposite Trend (Upregulated in IBD Enteroids and Downregulated in IBD Colonoids and Vice Versa) between IBD Enteroids and Colonoids Following LPS Stimulation				
Probe Set ID	Gene Description	Gene Symbol	LPS Treated IBD Colonoids vs. Control IBD Colonoids (Contrast 3) (log2 M)	LPS Treated IBD Enteroids vs. Control IBD Enteroids (Contrast 2) (log2 M)
Cfa.11069.1.A1_s_at	Splicing factor 3a subunit 1	*SF3A1*	0.39123732	−0.689723
Cfa.12226.1.A1_at	Double-headed protease inhibitor, submandibular gland	*LOC111092171*	0.13751692	−0.3472223
Cfa.6267.3.S1_s_at	Cysteine rich protein 1	*CRIP1*	0.43971334	−0.3231234
Cfa.758.1.S1_at	Annexin A1	*ANXA1*	0.57352154	−0.2534974
Cfa.1341.1.S1_s_at	Integral membrane protein 2B	*ITM2B*	0.09784959	−0.2300856
Cfa.11085.1.A1_at	Regulator of G-protein signaling 2	*RGS2*	0.40545754	−0.2130524
Cfa.3290.1.S1_at	Kruppel-like factor 6	*KLF6*	0.50830344	−0.2017464
CfaAffx.22128.1.S1_at	S100 calcium binding protein P	*S100P*	0.17803743	−0.1172426
Cfa.5123.1.A1_at	Eukaryotic translation initiation factor 3, subunit F	*EIF3F*	0.20471796	−0.1131341
Cfa.14421.1.S1_at	Guanine nucleotide binding protein (G protein), gamma 5	*GNG5*	0.22490961	−0.1044874
Cfa.415.1.S1_at	Catenin (cadherin-associated protein), beta 1	*CTNNB1*	0.09870051	−0.0672268
CfaAffx.5204.1.S1_s_at	Cytochrome c, somatic	*CYCS*	−0.3244678	0.117936
CfaAffx.17908.1.S1_s_at	Mitochondrial import receptor subunit TOM20 homolog	*TOMM20*	−0.6786731	0.18335565
Cfa.9946.1.S1_at	General transcription factor IIA, 2	*GTF2A2*	−0.4167318	0.19831684
CfaAffx.727.1.S1_x_at	FAM168A, family with sequence similarity 168, member A	*FAM168A*	−0.4299426	0.20356136
CfaAffx.16356.1.S1_s_at	LOC608756, similar to 60 kDa heat shock protein, mitochondrial precursor (Hsp60) (60 kDa chaperonin) (CPN60) (Heat shock protein 60) (HSP-60) (Mitochondrial matrix protein P1) (P60 lymphocyte protein) (HuCHA60)	*HSPD1*	−0.5398836	0.21506239
Cfa.1128.1.S1_at	KPNA2, karyopherin alpha 2 (RAG cohort 1, importin alpha 1)	*KPNA2*	−0.2790694	0.24160302
Cfa.6212.1.A1_at	Mitochondrial 28S ribosomal protein S21	*MRPS21*	−0.5480801	0.40259764
Cfa.12820.2.A1_a_at	Cysteine-rich, angiogenic inducer, 61	*CYR61*	−0.574112	0.60011614
CfaAffx.386.1.S1_x_at	40S Ribosomal protein S20	*RPS20*	−0.7323368	0.70095917
**Gene Ontology (GO) Analysis of DEGs (Annotated for Biological Process)**				
**GO-Terms**		**Genes**		
Cellular metabolic process (14%)		*RPS20*, *CYCS*, *EIF3F*, *MRPS21*, *LOC111092171*, *ANXA1*, *ITM2B*, *GTF2A2*, *SF3A1*, *HSPD1*		
Primary (nitrogen compound, organic substance) metabolic process (13%)		*RPS20*, *EIF3F*, *MRPS21*, *LOC111092171*, *ANXA1*, *ITM2B*, *GTF2A2*, *SF3A1*, *HSPD1*		
Biosynthetic process (7%)		*RPS20*, *EIF3F*, *MRPS21*, *ITM2B*, *GTF2A2*		
Regulation of cellular process (7%)		*LOC111092171*, *ANXA1*, *GNG5*, *ITM2B*, *HSPD1*		
Regulation of metabolic process/negative regulation of cellular process (6%)		*LOC111092171*, *ANXA1*, *ITM2B*, *HSPD1*		
Macromolecule localization and establishment (4%)		*TOMM20*, *ANXA1*, *KPNA2*		
Cell communication, signal transduction, cellular response to stimulus (4%)		*ANXA1*, *GNG5*, *HSPD1*		
Cellular component organization or biogenesis (4%)		*TOMM20*, *ANXA1*, *SF3A1*		
Regulation of molecular function (4%)		*LOC111092171*, *ANXA1*, *HSPD1*		
Negative regulation of metabolic process (4%)		*LOC111092171*, *ANXA1*, *ITM2B*		
Cell death (4%)		*CYCS*, *ANXA1*, *HSPD1*		
Cell population proliferation, regulation of multicellular organismal process, cytokine production, leukocyte activation, regulation of biological quality, positive regulation of immune system process, response to stress, regulation of immune system process, immune effector process, regulation of response to stimulus, cell activation, immune response, cell adhesion, positive regulation of cellular process, multicellular organism development, positive regulation of multicellular organismal process, immune system development, positive regulation of metabolic process, anatomical structure development (3%)		*ANXA1*, *HSPD1*		
Regulation of hydrolase activity (3%)		*LOC111092171*, *HSPD1*		
Response to chemical, positive regulation of response to stimulus (3%)		*ANXA1*, *HSPD1*		
Negative regulation of catalytic activity (3%)		*LOC111092171*, *ANXA1*		
Cellular localization (3%)		*TOMM20*, *KPNA2*		
Transmembrane transport (1%)		*TOMM20*		
Response to endogenous stimulus/external stimulus, negative regulation of transport/response to stimulus/immune system process/multicellular organismal process (1%)		*ANXA1*		
Cell motility, tissue migration, movement of cell or subcellular component, anatomical structure formation involved in morphogenesis, actin filament-based process, cell cycle process, cell–cell signaling (1%)		*ANXA1*		
Developmental growth, cellular developmental process, export from cell, Vesicle-mediated transport, positive regulation of locomotion/transport (1%)		*ANXA1*		
Taxis, localization of cell, leukocyte migration/homeostasis, myeloid cell homeostasis (1%)		*ANXA1*		
Regulation of developmental process/localization/locomotion/signaling, positive/negative regulation of developmental process (1%)		*ANXA1*		
Oxidation–reduction process (1%)		*CYCS*		
Production of molecular mediator of immune response, somatic diversification of immune receptors (1%)		*HSPD1*		
Protein folding, positive regulation of catalytic activity (1%)		*HSPD1*		
Biological process involved in symbiotic interaction, response to abiotic/biotic stimulus (1%)		*HSPD1*		
**Gene Ontology (GO) Analysis of DEGs (Annotated for Molecular Function)**				
**GO-Terms**		**Genes**		
Nucleic acid binding (15%)		*RPS20*, *EIF3F*, *SF3A1*, *HSPD1*		
Cation binding (12%)		*CYCS*, *ANXA1*, *S100P*		
Enzyme inhibitor activity (8%)		*LOC111092171*, *ANXA1*		
Nucleotide/ribonucleotide/nucleoside phosphate/anion binding (8%)		*ITM2B*, *HSPD1*		
Peptide binding (8%)		*TOMM20*, *ITM2B*		
Ion transmembrane transporter activity (4%)		*TOMM20*		
Phospholipid binding, calcium-dependent protein binding (4%)		*ANXA1*		
Electron transfer activity, tetrapyrrole binding (4%)		*CYCS*		
Translation initiation factor binding, translation factor activity, RNA binding (4%)		*EIF3F*		
G-protein beta-subunit binding (4%)		*GNG5*		
Lipopolysaccharide/apolipoprotein binding (4%)		*HSPD1*		
Protein–lipid complex binding, enzyme binding, chaperone binding (4%)		*HSPD1*		
p53 binding (4%)		*HSPD1*		
Nuclear import signal receptor activity (4%)		*KPNA2*		
Peptidase regulator activity (4%)		*LOC111092171*		
Signaling receptor binding (4%)		*S100P*		
Active/amide/protein transmembrane transporter activity (4%)		*TOMM20*		
Unfolded protein binding (4%)		*TOMM20*		
**Gene Ontology (GO) Analysis of DEGs (Annotated for Cellular Component)**				
**GO-Terms**		**Genes**		
Intracellular organelle (16%)		*RPS20*, *TOMM20*, *CYCS*, *MRPS21*, *ANXA1*, *ITM2B*, *GTF2A2*, *SF3A1*, *HSPD1*		
Membrane-bounded organelle (13%)		*TOMM20*, *CYCS*, *ANXA1*, *ITM2B*, *GTF2A2*, *SF3A1*, *HSPD1*		
Intracellular organelle lumen (7%)		*CYCS*, *ANXA1*, *GTF2A2*, *HSPD1*		
Plasma membrane (7%)		*ANXA1*, *ITM2B*, *GNG5*, *HSPD1*		
Mitochondrion (5%)		*TOMM20*, *CYCS*, *HSPD1*		
Intracellular non-membrane-bounded organelle (5%)		*RPS20*, *MRPS21*, *ANXA1*		
Intracellular vesicle, cytoplasmic vesicle (5%)		*ANXA1*, *ITM2B*, *HSPD1*		
Integral component of membrane (5%)		*TOMM20*, *ITM2B*, *HSPD1*		
Organelle envelope (4%)		*TOMM20*, *CYCS*		
Organelle membrane (4%)		*TOMM20*, *ITM2B*		
Extracellular organelle, plasma membrane region, endosome, extracellular exosome (4%)		*ANXA1*, *HSPD1*		
Intrinsic component of organelle membrane (4%)		*TOMM20*, *ITM2B*		
Extrinsic component of plasma membrane (4%)		*ANXA1*, *GNG5*		
Plasma membrane bounded cell projection, cilium, apical plasma membrane, external side of plasma membrane (2%)		*ANXA1*		
Supramolecular polymer (2%)		*ANXA1*		
Plasma membrane protein complex, extrinsic component of cytoplasmic side of plasma membrane, GTPase complex (2%)		*GNG5*		
Nuclear DNA-directed RNA polymerase complex, RNA polymerase complex, RNA polymerase II (holoenzyme), RNA polymerase II transcription regulator complex, transferase complex (2%)		*GTF2A2*		
Lipopolysaccharide receptor complex, secretory granule, Clathrin-coated pit (2%)		*HSPD1*		
Golgi apparatus (2%)		*ITM2B*		
Ribosomal subunit (2%)		*RPS20*		
Spliceosomal complex, small nuclear ribonucleoprotein complex (2%)		*SF3A1*		
Outer membrane, Mitochondria-associated endoplasmic reticulum membrane, outer mitochondrial membrane protein complex (2%)		*TOMM20*		

**Table 14 cancers-14-03525-t014:** The list of genes in the same or the opposite direction (upregulated in IBD enteriods/colonoids and downregulated in tumor-adjacent organoids and vice versa) between the IBD enteroids/colonoids and tumor-adjacent organoids following LPS stimulation. The log-ratio *M* values represent the log(R/G) (log fold change) [44].

Probe Set ID	Gene Description	Gene Symbol	LPS Treated vs. Control Tumor-Adjacent Organoids (Log-Ratio *M*)	LPS Treated vs. Control IBD Colonoids (Log-Ratio *M*)	LPS Treated vs. Control IBD Enteroids (Log-Ratio *M*)
Cfa.6267.3.S1_s_at	Cysteine rich protein 1 (CRIP1)	*CRIP1*	−0.238149286	0.43971334	−0.3231234
Cfa.758.1.S1_at	Annexin A1	*ANXA1*	0.309584226	0.57352154	−0.2534974
Cfa.11085.1.A1_at	Regulator of G-protein signaling 2	*RGS2*	−0.302506821	0.40545754	−0.2130524

**Table 15 cancers-14-03525-t015:** The KEGG enrichment analysis results by Blast2GO [48]. The details of each KEGG pathway [45,46,47] are available at https://www.kegg.jp/kegg/kegg1.html. * Top upregulated and downregulated genes were studied by Blast2GO [48].

Group Comparison *	Probe Set ID	Symbol	Name	Pathway
LPS Treated IBD sunek Colonoids vs. Control IBD Colonoids	CfaAffx.13209.1.S1_s_at	*CP*	Ceruloplasmin [EC:1.16.3.1]	Porphyrin metabolism, ferroptosis
	Cfa.6996.1.A1_at	*ENPP6*	Ectonucleotide pyrophosphatase/phosphodiesterase family member 6 [EC:3.1.4.-]	Ether lipid metabolism
Control IBD Colonoids vs. Control IBD Enteroids	CfaAffx.13785.1.S1_at, Cfa.6413.1.A1_at	*CA1*	Carbonic anhydrase [EC:4.2.1.1]	Nitrogen metabolism
	Cfa.20798.1.S1_at, Cfa.3774.1.A1_s_at	*ANPEP (LOC112653425)*	Aminopeptidase N [EC:3.4.11.2]	Glutathione metabolism, renin–angiotensin system, hematopoietic cell lineage
	CfaAffx.4748.1.S1_at	*BAAT*	Bile acid-CoA:amino acid N-acyltransferase [EC:2.3.1.65 3.1.2.2]	Primary bile acid biosynthesis, taurine and hypotaurine metabolism, Biosynthesis of unsaturated fatty acids, Peroxisome, Bile secretion
Control IBD Enteroids vs. Control Tumor Enteroids	CfaAffx.13785.1.S1_at, Cfa.6413.1.A1_at	*CA1*	Carbonic anhydrase [EC:4.2.1.1]	Nitrogen metabolism
	Cfa.20798.1.S1_at, Cfa.3774.1.A1_s_at	*ANPEP (LOC112653425)*	Aminopeptidase N [EC:3.4.11.2]	Glutathione metabolism, renin–angiotensin system, hematopoietic cell lineage
	Cfa.20396.1.S1_at	*BACE2*	Beta-site APP-cleaving enzyme 2 (memapsin 1) [EC:3.4.23.45]	Alzheimer’s disease
	CfaAffx.4748.1.S1_at	*BAAT*	Bile acid-CoA:amino acid N-acyltransferase [EC:2.3.1.65 3.1.2.2]	Primary bile acid biosynthesis, taurine and hypotaurine metabolism, biosynthesis of unsaturated fatty acids, peroxisome, bile secretion
LPS treated IBD colonoids vs. LPS treated IBD enteroids	CfaAffx.13785.1.S1_at, Cfa.6413.1.A1_at	*CA1*	Carbonic anhydrase [EC:4.2.1.1]	Nitrogen metabolism
	Cfa.18640.1.S1_at	*ACSL5*	Acyl-CoA synthetase long chain family member 5 [EC:6.2.1.3]	Fatty acid biosynthesis, fatty acid degradation, fatty acid metabolism, PPAR signaling pathway, ferroptosis, Thermogenesis, adipocytokine signaling pathway
	Cfa.6996.1.A1_at	*ENPP6*	Ectonucleotide pyrophosphatase/phosphodiesterase family member 6 [EC:3.1.4.-]	Ether lipid metabolism
	CfaAffx.4748.1.S1_at	*BAAT*	Bile acid-CoA:amino acid N-acyltransferase [EC:2.3.1.65 3.1.2.2]	Primary bile acid biosynthesis, taurine and hypotaurine metabolism, biosynthesis of unsaturated fatty acids, peroxisome, bile secretion
LPS treated IBD enteroids vs. LPS treated tumor enteroids	CfaAffx.13785.1.S1_at, Cfa.6413.1.A1_at	*CA1*	Carbonic anhydrase [EC:4.2.1.1]	Nitrogen metabolism
	Cfa.18640.1.S1_at	*ACSL5*	Acyl-CoA synthetase long chain family member 5 [EC:6.2.1.3]	Fatty acid biosynthesis, fatty acid degradation, fatty acid metabolism, PPAR signaling pathway, ferroptosis, thermogenesis, adipocytokine signaling pathway
	Cfa.6996.1.A1_at	*ENPP6*	Ectonucleotide pyrophosphatase/phosphodiesterase family member 6 [EC:3.1.4.-]	Ether lipid metabolism
	CfaAffx.4748.1.S1_at	*BAAT*	Bile acid-CoA:amino acid N-acyltransferase [EC:2.3.1.65 3.1.2.2]	Primary bile acid biosynthesis, taurine and hypotaurine metabolism, biosynthesis of unsaturated fatty acids, peroxisome, bile secretion
Common genes between IBD enteroids and colonoids following LPS stimulation	Cfa.2878.1.A1_s_at	*CP*	Ceruloplasmin [EC:1.16.3.1]	Porphyrin metabolism, ferroptosis
	CfaAffx.21951.1.S1_s_at	*EEF1A1*	Eukaryotic translation elongation factor 1 alpha 1 (eEF1A1)	Nucleocytoplasmic transport, legionellosis, leishmaniasis
	CfaAffx.2163.1.S1_at	*TAP2 (ABCB3)*	ATP-binding cassette, subfamily B (MDR/TAP), member 3 [EC:7.4.2.14]	ABC transporters, phagosome, antigen processing and presentation, human cytomegalovirus infection, Herpes simplex virus 1 infection, Epstein–Barr virus infection, human immunodeficiency virus 1 infection, primary immunodeficiency
	CfaAffx.17868.1.S1_at	*GPX1*	Glutathione peroxidase 1 [EC 1.11.1.9]	Glutathione metabolism, arachidonic acid metabolism

## Data Availability

The data presented in this study are available in the Appendix A.

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
