# Peer review of "Differential Transcriptomic Profiles Following Stimulation with Lipopolysaccharide in Intestinal Organoids from Dogs with Inflammatory Bowel Disease and Intestinal Mast Cell Tumor"

_cancers, 2022, doi:10.3390/cancers14143525_

Round 1
Reviewer 1 Report
This paper reports differential transcriptomes of LPS on canine enteroids and colonoids with IBD or intestinal tumor. With comprehensive comparisons among different groups, various upregulated and downregulated genes are here presented.
1. Figure 11, Acyl-CoA synthetase 5 (ACSL5) is highly expressed in intestine and is a mitochondrially localized enzyme, it is not localized in peroxisomes (Christina Klaus, et al. TP53 status regulates ACSL5-induced expression of mitochondrial mortalin in enterocytes and colorectal adenocarcinomas. Cell Tissue Res 2014 Jul;357(1):267-78).
If so, authors should add a reference for peroxisomal enzyme ACSL5.
2. Line 164, please indicate the pathogen where LPS was prepared from, E.Coli ?. LPS was obtained from where ? The preparation of LPS includes vortexing or sonication ?
3. Figure 1c, 4 labels on x-axis are scrambled and difficult to read. Change orientation to slanted labels for better viewing.
4. Figure 1d, how are ‘crypt’ and the length of ‘crypt’ defined ?
Author Response
We are grateful to the editor and reviewers for providing us with informative and helpful comments and ideas that have allowed us to improve the overall quality of our manuscript. We addressed all the reviewers' and editor's concerns. All amendments to our manuscript are noted in yellow.
Comments and Suggestions for Authors
This paper reports differential transcriptomes of LPS on canine enteroids and colonoids with IBD or intestinal tumor. With comprehensive comparisons among different groups, various upregulated and downregulated genes are here presented.
- Figure 11, Acyl-CoA synthetase 5 (ACSL5) is highly expressed in intestine and is a mitochondrially localized enzyme, it is not localized in peroxisomes (Christina Klaus, et al. TP53 status regulates ACSL5-induced expression of mitochondrial mortalin in enterocytes and colorectal adenocarcinomas. Cell Tissue Res 2014 Jul;357(1):267-78).
If so, authors should add a reference for peroxisomal enzyme ACSL5.
Response: We agree with the suggestion, and we modified Figure 11 (on page 85 of the revised manuscript) to remove ACSL5 from peroxisome localization.
- Line 164, please indicate the pathogen where LPS was prepared from, E.Coli ?. LPS was obtained from where ? The preparation of LPS includes vortexing or sonication ?
Response: This has been addressed in the revised manuscript on page 9. Details are given below:
The LPS from Escherichia coli 026:B6 in an aqueous solution obtained from Thermo Fisher Scientific (Waltham, USA) was used for the study.
- Figure 1c, 4 labels on x-axis are scrambled and difficult to read. Change orientation to slanted labels for better viewing.
Response: This has been addressed in the revised manuscript on page 14.
- Figure 1d, how are ‘crypt’ and the length of ‘crypt’ defined?
Response: This has been addressed in the Discussion section of our manuscript revision on page 90. Details are given below:
Adult stem cells expressing the Leucine-rich repeat-containing G-protein coupled receptor (LGR5) can be identified near the base of intestinal crypts [64] that are employed for organoid proliferation. Using canine-specific LGR5 probes, we previously detected ISCs within intestine organoid crypts [30], demonstrating that proliferation is restricted to crypt-like budding structures [64].
Reviewer 2 Report
This is an interesting article concerning differential transcriptomic profiles following stimulation with LPS in Intestinal Organoids.
The topic is interesting and fashionable. The authors have well developed the methodology.
I think the clinical implications of the present study must be highlighted and a better introduction of the role of mast cells should be developed considering the involvement in several physiological and pathological processes
SakalauskaitÄ— S et al . Association of mast cell density, microvascular density and endothelial area with clinicopathological parameters and prognosis in canine mammary gland carcinomas. Acta Vet Scand. 2022 Jun 27;64(1):14. doi: 10.1186/s13028-022-00633-2
Sammarco et al. Mast Cells, microRNAs and Others: The Role of Translational Research on Colorectal Cancer in the Forthcoming Era of Precision Medicine. J Clin Med. 2020 Sep 3;9(9):2852. doi: 10.3390/jcm9092852
The discussion must be shortened (2 pages and not 4).
The limitations of the study should be better discussed
Author Response
We are grateful to the editor and reviewers for providing us with informative and helpful comments and ideas that have allowed us to improve the overall quality of our manuscript. We addressed all the reviewers' and editor's concerns. All amendments to our manuscript are noted in yellow.
Comments and Suggestions for Authors
This is an interesting article concerning differential transcriptomic profiles following stimulation with LPS in Intestinal Organoids.
The topic is interesting and fashionable. The authors have well developed the methodology.
I think the clinical implications of the present study must be highlighted and a better introduction of the role of mast cells should be developed considering the involvement in several physiological and pathological processes
SakalauskaitÄ— S et al . Association of mast cell density, microvascular density and endothelial area with clinicopathological parameters and prognosis in canine mammary gland carcinomas. Acta Vet Scand. 2022 Jun 27;64(1):14. doi: 10.1186/s13028-022-00633-2
Sammarco et al. Mast Cells, microRNAs and Others: The Role of Translational Research on Colorectal Cancer in the Forthcoming Era of Precision Medicine. J Clin Med. 2020 Sep 3;9(9):2852. doi: 10.3390/jcm9092852
Response:
This has been addressed in the Introduction section of our manuscript revision on page 4. The reference section is modified to include SakalauskaitÄ— et al. (2022) and Sammarco et al. (2020) as per the suggestions. Details are given below:
Mast cells (MCs) play an essential immunoregulatory role, especially at the mucosal barrier between the body and the environment. In intestinal lesions, MCs can be seen infiltrating the inflammatory microenvironment and regulating the synthesis of several pro-inflammatory cytokines and mediators of inflammatory cell production, thus promoting tumor growth and proliferation [14]. Numerous animal and human studies indicate the presence of an abundance of MCs surrounding tumor cells or inflammation lesions [14–17]. For instance, high MC density is correlated with advanced stage and tumor progression in colorectal cancer (CRC) [17]. Similarly, intratumoral MC density correlates with tumor size, and peritumoral MC density correlates with malignancy grade [16]. MCs also play a crucial part in IBD, and research has shown that the number of MCs increases in inflammatory bowel lesions in patients with IBD [14].
The discussion must be shortened (2 pages and not 4).
Response:
We understand the reviewer’s comment. However, we believe the Discussion section could not be shortened without removing key information that is critical to the understanding of our results, including all genes and their role in various signaling pathways associated with IBD and cancer development. We hope this is acceptable to the referee.
The limitations of the study should be better discussed.
Response:
This has been addressed in the Discussion section of our manuscript revision on page 98. Details are given below:
Organoids are confined to mimic organ-specific or tissue-specific micro-physiology, a constraint that should be considered before joining this promising new field. The lack of inter-organ communication is a major weakness of organoid systems. On the other hand, initiatives are already underway to circumvent this constraint. For instance, multiple organoids have been linked together to explore the gastrointestinal tract-liver-pancreas connection [165]. Despite the lingering difficulties, organoids retain significant promise for clinical translational research [166]. The scope of organoid technology has extended to include genetic manipulation, diverse omics and drug-screening studies, and a diversified co-culture system involving viruses, bacteria, and parasites. In the near future, the combination of organoid technology with single-cell transcriptomics will have a significant impact on personalized medicine.
Round 2
Reviewer 2 Report
I'm satisfied with the changes made